# SCALABLE MULTI-AGENT AUTONOMOUS LEARNING IN COMPLEX UNPREDICTABLE ENVIRONMENTS

## ABSTRACT

This research introduces a novel multi-agent self-learning solution for large and complex tasks in dynamic and unpredictable environments where large groups of homogeneous agents coordinate to achieve collective goals. Using a novel iterative two-phase multi-agent reinforcement learning approach, agents continuously learn and evolve in performing the task. In phase one, agents collaboratively determine an effective global task distribution based on the current state of the task and assign the most suitable agent to each activity. In phase two, the selected agent refines activity execution using a shared policy from a policy bank, built from collective past experiences. Merging agent trajectories across similar agents using a novel shared experience learning mechanism enables continuous adaptation, while iterating through these two phases significantly reduces coordination overhead. This novel approach was tested with an exemplary test system comprising drones, with results including real-world scenarios in domains like forest firefighting. This approach performed well by evolving autonomously in new environments with a large number of agents. In adapting quickly to new and changing environments, this versatile approach provides a highly scalable foundation for many other applications tackling dynamic and hard-to-optimize domains that are not possible today.

## 1 INTRODUCTION

Many real-world problems are quite big and complex, requiring many agents with different capabilities to effectively tackle them. Autonomous multi-agent applications like delivery systems, warehouse robots, and drone shows work in mostly deterministic and constrained environments. However, there are many complicated dynamic environments, such as forest fire-fighting, disaster relief, urban fire, and medical rescue operations involving collaboration between a very large number of agents, where each episode is unique and ridden with unpredictable challenges. Today's MARL algorithms fail to address the enormity and complexity of these tasks (Rashid et al., 2018) (Yu et al., 2022).

We propose a novel two-phase iterative approach to enable groups of homogeneous agents with different capabilities to autonomously learn to perform huge, unpredictable, fast-changing tasks. Phase One - Refocus: determines the best way to target the task, Phase Two: Refine - uses the collective intelligence of the group for each agent to best perform its task, and iteratively repeating this leads to continuous evolution. This opens the possibility of complementing pure reinforcement learning with adjunct strategies, including domain intelligence or human-in-the-loop (HIL), to expedite learning. It realigns learning to focus on the most relevant portion of the state-space and gives agents autonomy to improvise while significantly reducing the coordination effort across numerous agents. Using shared experience across homogeneous agents with a shared population policy bank, this result-oriented learning is highly scalable. We demonstrate this approach via an exemplary test system comprising drones fighting forest fires.

## 2 RELATED WORK

Recent progress in multi-agent reinforcement learning (MARL) has enabled significant achievements in complex environments, yet scaling up to large, dynamic, and unpredictable tasks remains challenging. Scalability issues arise due to exponential growth in state space and agent interactions, along with multi-agent variance and multi-observation variance (Hopkins, 2024). With partial observability,

non-stationarity, and dynamic environments, these significantly hinder stable learning, (Wei et al., 2024; Liang et al., 2025) underscoring the need for improved frameworks that can handle large-scale multi-agent coordination more efficiently.

One approach to manage large problems is to adopt hierarchical reinforcement learning (HRL). Hierarchical RL techniques reduce dimensionality by decomposing tasks into subtasks governed by high-level policies (Dietterich, 2000; Levy et al., 2019; Nachum et al., 2019). These methods define high-level policies that operate over temporally extended actions or subtasks, thereby pruning the search space. However, reliance on static decompositions or domain knowledge limits their applicability when tasks evolve significantly over time (e.g., rapidly shifting operational zones). Additionally, current approaches for subgoal discovery (Pateria et al., 2022) (Wang et al., 2025), learning when to retrain (Haighton et al., 2023), and learning hierarchical world models (Schiewer et al., 2024) could have limited scalability for large-scale tasks involving a large number of agents.

Dividing large tasks into subtasks and assigning them to homogeneous agents is combinatorial and non-trivial, often leading to overlapping roles or inefficient exploration (Martins et al., 2025; Zheng et al., 2018). Repetitive subtasks (e.g., scouting or delivery) can be addressed through a policy bank of pre-optimized solutions (Teh et al., 2017; Rusu et al., 2016), enabling faster adaptation. Joint experience-sharing—via parameter, memory, or replay sharing—further improves learning efficiency (Gupta et al., 2017; Rashid et al., 2018). Nonetheless, scaling these techniques to truly massive and fluid domains remains a key research challenge. Some collaborative MARL approaches perform role assignment by matching latent subtask representations with latent trajectory representations and use algorithms like QMIX to mix similar policies. However, this approach limits scalability and limits expressivity for activities and constraints. (Yang et al., 2022) (Xia et al., 2023) (You et al., 2025) Automated grouping approaches (Zang et al., 2023) and role assignment (Nguyen et al., 2022) can also limit expressivity and scalability. Here we address the large, fast-changing state-space aided by a task-specific means to decompose an activity assignment and use a policy bank to address many types of activities that are still commonplace for the huge tasks, and learn these policies through shared experiences of homogeneous agents.

## 3 PROPOSED APPROACH

### 3.1 TASK DECOMPOSITION, ASSIGNMENT, AND EXECUTION POLICY

Consider a dynamic task $\mathcal{W}(t)$ that evolves over time $t$. The task is performed by a set of $N$ agents partitioned into $G$ homogeneous groups, such that $\mathcal{A} = \bigcup_{g=1}^{G} \mathcal{A}_g$. Each group $\mathcal{A}_g$ consists of agents with identical capabilities, meaning a minimum set of capabilities $\mathcal{C}_g = \mathcal{C}(a_{gi})$ for all $a_{gi} \in \mathcal{A}_g$. The task $\mathcal{W}(t)$ is composed of $M_t$ activities, where $\mathcal{W}(t) = \{w_{1t}, w_{2t}, \ldots, w_{M_t t}\}$.

Each activity $w_{jt}$ has an associated complexity level $c(w_{jt})$ and requires capabilities $\mathcal{C}(w_{jt})$. The activities change over time, appearing and disappearing based on the task's evolving state. Each activity $w_{jt}$ has an associated *relevance duration* $(t_j^{\text{start}}, t_j^{\text{end}})$ such that the activity exists within the time window $t_j^{\text{start}} \leq t \leq t_j^{\text{end}}$. Activities dynamically emerge and vanish depending on task conditions. The presence of an activity is determined by the function $\Psi(\mathcal{W}(t), t)$, such that $w_{jt}$ where $1 \leq j \leq M_t$ exists at time $t$ if $\Psi(\mathcal{W}(t), t) = 1$. The task state function $\Phi(\mathcal{W}(t))$ describes the current status of the task and influences which activities are required.

Since agents in a group share capabilities, task decomposition ensures that there are multiple similar activities to fully utilize homogeneous agents. A decomposition function $\mathcal{D}$ partitions the task into activities that match group capabilities, i.e., $\mathcal{D}(\mathcal{W}(t)) = \bigcup_{g=1}^{G} \mathcal{W}_g(t)$, where $\mathcal{W}_g(t)$ is the subset of activities assigned to $\mathcal{A}_g$. Each activity $w_{jt} \in \mathcal{W}_g(t)$ must satisfy $\mathcal{C}(w_{jt}) \subseteq \mathcal{C}_g$. The decomposition process aims to generate enough similar activities such that $|\mathcal{W}_g(t)| \geq |\mathcal{A}_g|$, for full agent utilization.

Each agent $a_{gi} \in \mathcal{A}_g$ is assigned an activity from $\mathcal{W}_g(t)$. The binary assignment matrix $X_t \in \{0,1\}^{N \times M_t}$ is defined such that $x_{ij} = 1$ if agent $a_i$ is assigned to activity $w_{jt}$, otherwise $x_{ij} = 0$. This assignment of an agent to an activity can be optimized in many ways, depending on the overall goal of executing the task. This optimization directly impacts the efficacy of performing the task, and therefore, we formalize this framework here to be able to explore this issue in the subsequent sections. For example, if the goal is to perform the task so as to minimize the execution cost,

the agent assignment must minimize $\sum_{i=1}^{N} \sum_{j=1}^{M_t} C(a_i, w_{jt}) x_{ij}$, where $C(a_i, w_{jt})$ is the execution cost of agent $a_i$ working on activity $w_{jt}$. In general, it must do so while satisfying constraints $\sum_{i=1}^{N} x_{ij} \geq r(w_{jt})$, where $r(w_{jt})$ is the minimum number of agents required to execute activity $w_{jt}$. This ensures each activity is assigned sufficient agents. Moreover, $\sum_{j=1}^{M_t} x_{ij} \leq \kappa(a_i)$, where $\kappa(a_i)$ is the maximum number of activities that agent $a_i$ can handle at a given time, thereby ensuring to limit the agent's workload. A simplified representation of this agent assignment is a function $\mathcal{S}(w_{jt})$ that determines the set of agents executing $w_{jt}$, such that $\mathcal{S}(w_{jt}) = \{a_i \in \mathcal{A} | x_{ij} = 1\}$.

Various operational concerns, such as business, technical, and logistics, may determine a task decomposition and activity assignment to agents for many complex real-world tasks. Additionally, for the task to be optimized, in addition to an effective agent assignment, it is necessary to also ensure that each agent $a_i$ optimizes the execution of its assigned activity $w_{jt}$. A task performance metric is given by $J(\mathcal{W}(t), X_t, \Pi)$, where $\mathcal{W}(t)$ represents task activities at time $t$, $X_t$ is the agent-assignment matrix, and $\Pi$ denotes execution policies. The goal is to meet all operational concerns and also to continuously adapt $\mathcal{W}(t)$, $X_t$, and $\Pi$ such that $J$ improves over time. $\mathcal{W}^*(t)$ and $X_t^*$ are comprehensive when operational concerns are met and when $\sum_{i=1}^{N} x_{ij} \geq r(w_{jt})$ and $\sum_{j=1}^{M_t} x_{ij} \leq \kappa(a_i)$.

Task decomposition to match agent capabilities and generate balanced activities is a combinatorial problem that is often NP-hard. MARL algorithms struggle with such problems, particularly at scale, as shown in Gu et al. (2020); Martins et al. (2025). They rely on local rewards, perform poorly in discrete combinatorial spaces, and converge slowly in dynamic environments. By Bellman's principle of optimality, if task decomposition and assignment are suboptimal, as with MARL, overall task performance cannot be optimal with MARL.

### 3.2 TWO-PHASE APPROACH

Given MARL's limitations in optimally decomposing and assigning tasks in dynamic environments, we introduce a two-phase approach, as illustrated in Figure 1. This iterative process enables continuous adaptation to evolving tasks while ensuring that homogeneous agents execute activities using the most effective policies available.

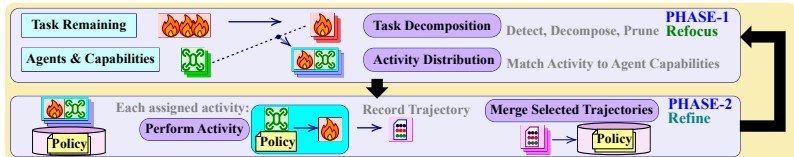

Figure 1: Two-phase approach - iterating between Phase-1 refocusing agent task distribution and Phase-2 executing activity with the best policy from the policy bank and shared experience merging of optimal trajectories leads to continuous learning.

### 3.2.1 PHASE ONE - TASK DECOMPOSITION AND ASSIGNMENT

In phase one, each agent helps obtain information from their environment and shares it with a task distributor. The task distributor decomposes the task in its current state into activities and distributes these activities to the most suitable agents. This allows segmenting a massive state space for a huge task into smaller-scale activities that agents can handle, as discussed in the last section. During phase-1, we optimize task decomposition $W^*(t)$ with an optimal assignment $X_t^*$ for all possible tasks $W(t)$ by matching the agent capabilities with the activities. This opens the possibility of complementing pure reinforcement learning with adjunct strategies by leveraging AI-driven task decomposition and assignment methods. Additionally, it becomes feasible to use domain-centric, oracle-centric, human-in-the-loop (HIL), another learning approach, or a combination of these approaches to aid in determining what is the best way for the group of agents to tackle the current state of the task.

At each timestep $t$, given the current task state $\Phi(W(t))$, we seek an optimal task decomposition $W^*(t)$ and assignment matrix $X_t^*$. As this is repeated regularly, the system adapts to the dynamic nature of the task and refocuses agents to operate on the currently most relevant aspects of the task. Unlike a pure MARL approach where each agent would learn to tackle a vast task state-space, phase

one could refocus the agents at every iteration of the two-phase approach to attend a specific, narrow state-space of an activity that is most likely to make an immediate and significant contribution to the overall task. By decomposing the large task environment into small activities that an agent can execute, it becomes amenable to optimization at the local level by the most appropriate reinforcement learning algorithms known for the activity.

### 3.2.2 Phase Two - Policy Execution and Learning

Each agent $a_i \in \mathcal{A}_g$ selects the best-known policy for its assigned activity $w_{jt}$ from a policy bank. Agents execute activities using RL/MARL algorithms such as PPO and record trajectories. A merge operation refines the policy based on the best experiences and stores the updated policy back into the policy bank.

Each agent $a_i \in \mathcal{A}_g$ executes its assigned activity $w_{jt}$ using a policy from a policy bank $\mathcal{B}_g$, where $\pi_{w_{jt}} = \Pi(w_{jt}) \in \mathcal{B}_g$. The policy $\pi_{w_{jt}}$ is selected based on the similarity between the assigned activity and previously encountered activities. At each time step $\tau$, an agent $a_i \in \mathcal{A}_g$ selects an action $a_\tau \sim \pi_{w_{jt}}(h_\tau)$. The policy $\pi_{w_{jt}} : H \times A \to [0, 1]$, where $H$ represents histories and $A$ represents agents. Agents with same capabilities belong to a group $A_g$, and these homogeneous agents share their experiences to collectively refine $\pi_{w_{jt}}$.

Consider $\mathcal{A}_g$ agents executing activity $w_{jt}$, each following initial policy $\pi_{w_{jt}}$, with expected policy performance $J(\pi_{w_{jt}}) = \mathbb{E}_{\zeta \sim \pi_{w_{jt}}}[R(\zeta)]$, where $R(\zeta)$ is the expected return over trajectory $\zeta$. Each agent $a_i$ collects experience $\mathcal{E}_{a_i} = \{(o_\tau, a_\tau, r_\tau, o_{\tau+1}) \mid \tau = 0, \ldots, T_\zeta\}$ for $T_\zeta$ trajectory samples, with POMDP observations $o \in O$, actions $a \in A$, reward $r \in \mathbb{R}$, and timestep $\tau$. The policy improvement in $\pi_{w_{jt}}$ after $k$ updates for experience distribution $\mathcal{E}$ is given by $J(\pi_{w_{jt}}^{(k)}) = J(\pi_{w_{jt}}^{(k-1)}) + \alpha \mathbb{E}_{(o,a) \sim \mathcal{E}}[\nabla J(\pi_{w_{jt}})]$. For individual learning, $\mathcal{E} = \mathcal{E}_{a_i}$. With a merge strategy $\mathcal{M}$, shared learning aggregates experience as $\mathcal{E} = \mathcal{M}(\mathcal{E}_{a_1}, \mathcal{E}_{a_2}, \ldots, \mathcal{E}_{a_{|\mathcal{A}_g|}})$.

**Proposition 1.** [Convergence Acceleration via Merged Learning] If $p$ homogeneous agents merge the top and bottom $n$ % of the combined trajectories, the policy learns $2pn$ times faster than for a single agent learning using all its trajectories.

**Lemma 1.1.** [Policy Update through Experience Merging] Updating policy $\pi_{w_{jk}}$ through experience merging with the best and worst $n\%$ trajectories $\zeta$ across all homogeneous agents ensures improvement in expected task performance: $\mathbb{E}[J(W^*(t), X_t^*, \Pi'_{w_{jt}})] \geq \mathbb{E}[J(W^*(t), X_t^*, \Pi_{w_{jt}})]$

Thus, homogeneous agents can collectively refine a single policy by pooling experiences, leading to faster and more stable learning. During execution, each agent collects experience tuples $\mathcal{E}_{a_i} = \{(o_\tau, a_\tau, r_\tau, o_{\tau+1})\}$. A merge operation refines the policy based on the best-performing trajectories: $\pi'_{w_{jt}} = \mathcal{M}(\pi_{w_{jt}}, \mathcal{E}_{\text{best}})$, where $\mathcal{M}$ integrates high and low reward trajectories into the stored policy. The updated policy replaces the existing one in the policy bank: $\mathcal{B}_g[w_{jt}] \leftarrow \pi'_{w_{jt}}$. This ensures groups of **homogeneous** agents continually refine and reuse the best available policies for task execution under partial observability.

**Proposition 2.** [Two-Phase Task Optimization] Let $J(W(t), X_t, \mathcal{B}_g)$ be the task performance function, where $\mathcal{B}_g$ is the policy bank. The iterative execution of phase one and phase two ensures the task policy converges to an optimal solution as the iterations progress if

1. Task decomposition and assignment are comprehensive: $(W^*(t)$ and $X_t^*)$, and

2. Policy update through experience merging ensures improvement in expected task performance: $\mathbb{E}[J(W^*(t), X_t^*, \Pi'_{w_{jt}})] \geq \mathbb{E}[J(W^*(t), X_t^*, \Pi_{w_{jt}})]$

**Theorem 1** (Task Learning). If there is a dynamic task $\mathcal{W}(t)$ decomposed and assigned comprehensively as $(W^*(t), X_t^*)$ as described in section 3.1, the task $\mathcal{W}(t)$ can be effectively distributed and learned among agents $a_i \in \mathcal{A}$.

Algorithm 1 demonstrates the two-phase approach where agents obtain the activity from task distributor, perform the activity using operateAgent procedure using a reinforcement learning algorithm suitable for the optimal policy for the activity, and collect their experiences in $D_i$. The agents use a merge strategy to update the policy using Algorithm 2. The updateSharedLearning procedure updates the policy based on the reinforcement learning algorithm used by the agent.

---

**Algorithm 1** Population policy MARL for agent $a_i$

---

1: Initialize populations $\Pi^0(b)$ for all agent activity-types $b \in B$
2: **for** each task iteration $k = 1, 2, 3, \cdots$ **do**
3:     Obtain activity assignment $t_i$ from task distributor.
4:     Select optimal policy for $b = type(t_i)$ as $\Pi^k(b)$ from population.
5:     $\Pi_i^k(b) = \Pi^k(b)$
6:     $D_i = \text{operateAgent}(a_i, \Pi_i^k(b))$
7:     Prune $D_i$ using merge strategy
8:     policyMerge($D_i, \Pi_i^k(b), a_i$)
9: **end for**

---

**Algorithm 2** Merging learned policies - policyMerge

---

**Require:** $D_i$ set of trajectories $(h_i^t, a_i^t, r_i^t, h_i^{t+1})$, $\Pi_i^k(b)$ policy, $a_i$ agent identity
1: $D_{shared} = \bigcup_{i \in I, type(t_i)=b} D_i$
2: Await potentially contributing agents $i \in I$ with $type(t_i) = b$
3: $\Pi^k(b) = \text{updateSharedLearning}(\Pi^k(b), D_{shared})$
4: save $\Pi^k(b)$ to population.

---

## 3.3 EXEMPLARY SYSTEM

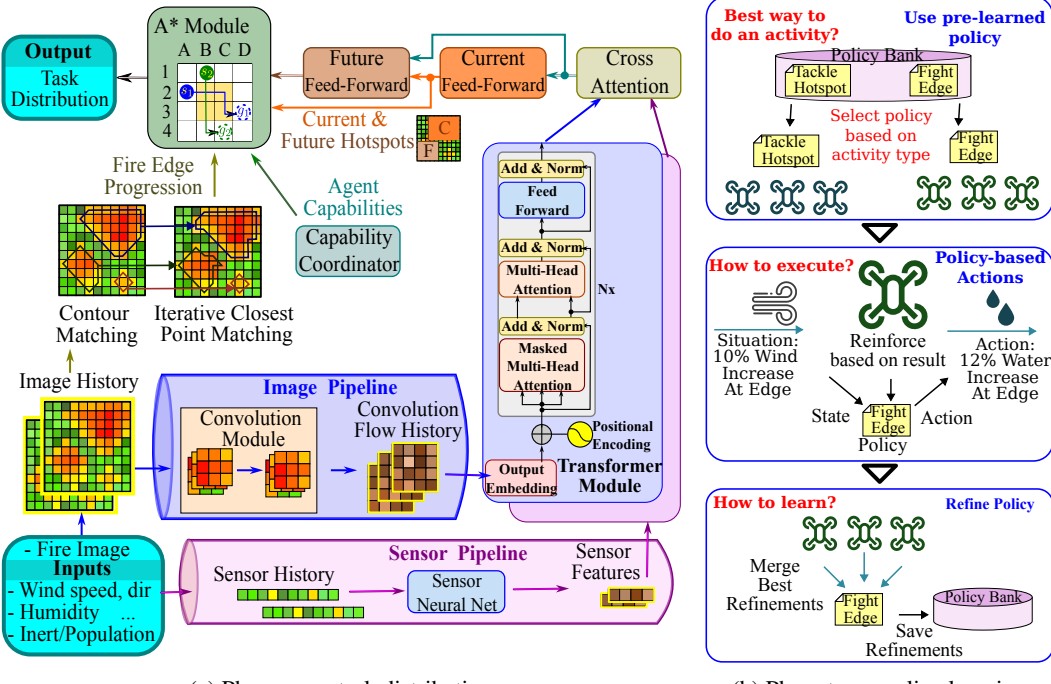

(a) Phase one - task distribution      (b) Phase two - policy learning

Figure 2: Exemplary two-phase approach for forest fire-fighting - standard Phase-1 task distribution complemented with forest fire fighting pipelines. Phase-2 activity execution with optimal policy selection followed by shared experience learning.

This approach was tested with an exemplary system as shown in Figures 2a and 2b. It works with a large number of simulated drones that can operate alongside a few actual replicas of real-world autonomous drones. Figure 2a shows phase one task distribution for a forest fire-fighting system used to showcase the implementation, testing, and results discussed here. This algorithm can handle task distribution for similar domains, such as flood control and synthetic domains. Here, the standard

phase-1 task distribution $A^*$ module and capability coordinator are complemented with fire-fighting specific pipelines to expedite learning.

Each drone takes in inputs of critical fire-fighting components like fire image, wind speed and direction, location, humidity, temperature, vegetation type, and population. It detects fire-spread locations and hotspots. An edge progression module detects fire boundary progression since the last time step. These components are fed into a convolution-transformer pipeline to detect current and predict future hotspots and their intensity. A task distributor collects the boundary and hotspot information along with drone capabilities and uses a heuristics-based $A^*$ planner to divide tasks and assign agents to an activity. Some exemplary activities include fight-edge and fight-hotspot of different sizes and intensities, as shown in Figure 2b.

Figure 2b shows phase two, where homogeneous agents $a_i \in \mathcal{A}_g$ select the best policy $\pi_{w_{jt}}$ from the policy bank for its activity $w_{jt}$. Each agent $a_i$ performs its activity $w_{jt}$ using RL/MARL algorithms based on PPO in Schulman et al. (2017), Actor-Critic in Konda & Tsitsiklis (2000), and DQN in Mnih et al. (2013) for $\pi_{w_{jt}}$. They gather their experience as in Algorithm 1 and merge their experiences as in Algorithm 2. Their shared experience evolves the system, and the two-phase approach allows adapting to the dynamics of forest fires in an effective manner.

## 4 RESULTS AND DISCUSSION

### 4.1 EXPERIMENTAL SETUP

This system was tested with the exemplary forest-fighting system disclosed in the last section. The simulation allows testing a large number of drones in a variety of simulated environments based on real fires. Testing with actual drones shows how the system can operate in the real world. A detailed description of the experimentation is disclosed in Appendix A.3Experiment Details.

Simulating wildfires is an active research area, with many accurate ways to model the fire and fire extinguishing. We used the WRF-Fire modeling guidelines in Coen et al. (2013) to determine the spread of wildfires based on factors like fuel and weather, and used Hansen (2012) to determine water extinguisher efficacy based on the spray angle, duration, and power, along with vegetation type. A custom simulator was created using these modeling guidelines to test our approach for fighting forest fires. A fleet of three custom-built drones that can coexist with more than 3000 simulated drones was used. The drones were built using a PixHawk with an Ardupilot flight controller, a LASER to emulate a fire extinguisher, and an onboard Raspberry Pi for autonomous operation in coordination with an on-ground custom ground controller integrated with the simulator.

The POMDP reward function $R_a$ used by agents is based on the change in fire intensity $\frac{\Delta I}{I}$ and fire-area $\frac{\Delta A}{A}$ as a result of an agent action. $R_a = \alpha \cdot min\left(\frac{\Delta I}{I}, k_1\right) + \beta \cdot min\left(\frac{\Delta A}{A}, k_2\right)$ where, factors $\alpha$ and $\beta$ control the weightage of changes in intensity and fire-area on the resulting reward. The experiments used by default $\alpha = 2500, \beta = 3500, k_1 = 0.02, k_2 = 0.02$ to balance the effects of both intensity and area. There is a slight overweight for area change, as a smaller area offers better opportunities to contain and fight with fewer high-capacity drones.

Both public datasets such as Singla et al. (2020); Fantineh (2023); Nguyen et al. (2024); Center (2025); NIFC (2025) and synthetic datasets using fire models were used for testing, to test specific aspects of the system for different fire scenarios. Fire was simulated with multi-colored fabric that can be moved along the ground, simulating different fire positions and intensities of a fire dataset sample. On-board drone CNN trained for this fabric fire simulation effectively helped simulate many fire scenarios. A fire unit represents a normalized unit area of full fire on the ground. Three homogeneous groups of drones with capability types small, medium, and large having speeds of 4x, 2x, and 1x and fire extinguisher capacities of 10 liters, 50 liters, and 100 liters respectively were used with varying density and fleet composition per fire unit.

A baseline of firefighters from the public datasets was used to evaluate the overall fire containment performance using fire containment time and extinguisher resources. The fire containment performance of 3,000 simulated agents - comprising a drone fleet with small:medium:large size ratios of 50:35:15 and equipped with water-based extinguishers - is compared to that of real firefighters. The evaluation focuses on improvements in containment time and efficiency of fire-extinguishing resource usage. This was tested for fires of different sizes and hotspots. To ensure repeatability and

consistency in performance, multiple trials were conducted to measure percentage improvements in time and resource usage across fires of different sizes. Specifically, medium fires with 10 hotspots and large fires with 30 hotspots were tested, each using 10 different random seeds.

The evaluation return for T timesteps is computed as the cumulative returns during multiple trial episodes, using the greedy policy after training it for T timesteps. The effect of individual components and algorithms on learning the policy is evaluated by comparing evaluation returns across configurations, as it isolates the learning dynamics of the training phase.

An ablation study of phase one components was done with a transformer, edge progression, and A* distributor to evaluate the efficacy of phase one and its components. The transformer was replaced by a no-transformer component that predicts the location and fire intensity using image analysis based on fire colors. The A* component was replaced by a rules-based distribution method, and the edge-progression component was replaced by a fire-edge contour detector along with the mean intensity along each contour.

Algorithms used for two-phase population policy-bank based learning are evaluated, including on-policy PPO, Advantage Actor-Critic, and off-policy DQN, and compared against traditional MARL versions of these algorithms with 25 agents, including MAPPO as in Yu et al. (2022), an A2C alternative of MAPPO, and QMIX as in Rashid et al. (2018). The scalability of this approach was examined by conducting a test, where a hotspot of the same size was assigned to each available agent and recording the total area fought in a fixed duration of 2000 timesteps.

The impact of trajectory merging based on shared experiences was analyzed in terms of the fire containment time improvement, while maintaining the same level of resource usage as under the fire-fighter baseline. Three trajectory merge strategies tested include Best-N, Hybrid-N, and Weighted-N trajectory merging. Their impact was evaluated using an ANOVA test for statistical significance. Trajectories from similar homogeneous agents were ranked based on reward and used for shared experience learning. The Best-N strategy merges the top N trajectories, hybrid-N merges the top and bottom N trajectories, and weighted N merges trajectories by repeating them multiple times based on their weights computed by their top and bottom ranks.

Note that we explored many standard benchmarks that exist for traditional MARL algorithms, such as the SMAC benchmark as in Samvelyan et al. (2019); Ellis et al. (2023) that focuses on zero-sum competitive games or games with a limited number of agents as discussed in Appendix A.5. These benchmarks did not allow evaluating the many aspects of our system for cooperative tasks with high scalability. Therefore, it was necessary to test this system with an exemplary firefighting system involving coordination between a large number of agents to cooperatively accomplish a complex, unpredictable, and fast-changing task like fighting forest fires.

## 4.2 COMPARISONS AND ANALYSIS

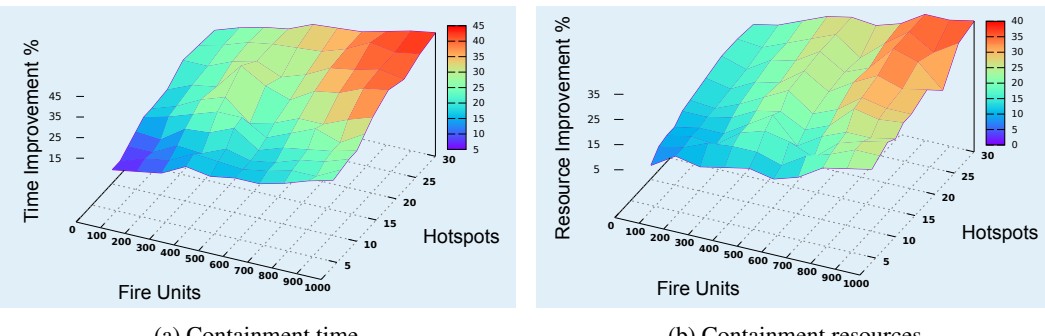

(a) Containment time                    (b) Containment resources

Figure 3: Containment performance

Figure 3 shows the fire containment time and resource improvement of our approach over the baseline system. In Figure 3a, regardless of the number of units and hotspots, our approach outperforms the baseline by over 15% and exceeds 40% for a large number of units and hotspots. In Figure 3b, as fire units and hotspots increase, our approach outperforms the baseline in resource consumption. As the

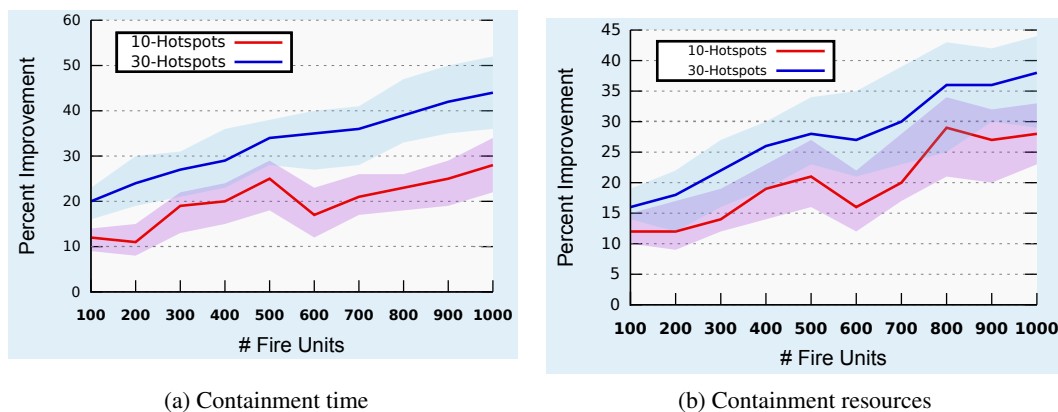

(a) Containment time            (b) Containment resources

Figure 4: Containment by hotspots

number of fire units and hotspots increases, optimizations along burning edges and hotspots increase, greatly reducing the containment time and fire-extinguishing resource usage. Figure 4a and 4b further support this observation, showing greater improvements with more hotspots and larger fire-sizes as bigger tasks offer more scope for optimizations.

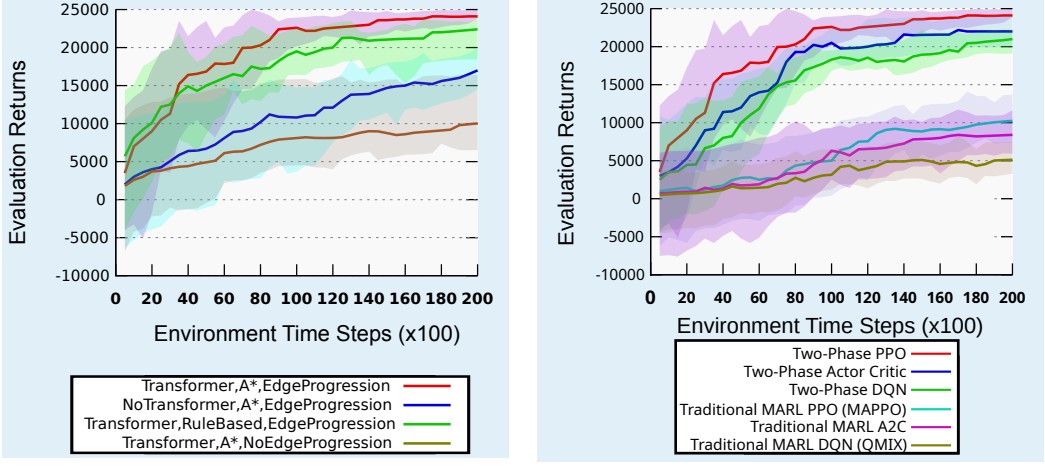

Figure 5: Task distribution methods        Figure 6: Two-phase learning algorithms

Figure 5 shows ablation study results with transformer, $A^*$, and edge progression providing the best performance, as edge progression helps detect edges, transformers predict hotspots, and $A^*$ best distributes when edges, hotspots, and agent capabilities are available. This also shows that task-specific combinatorial optimization quickly offers good performance. The transformer allows predicting future hotspots, and edge-progression allows accurate edge tracking, which are crucial in staying ahead of the fire spread by timely positioning and spraying the extinguisher. A* allows using heuristics based on danger quotient, which allows assigning high-capability drones to areas that are prone to maximum fire spread.

Figure 6 shows that phase two algorithms significantly outperform traditional MARL algorithms. Phase One refocuses training to relevant activities, and phase two uses the best known policies to efficiently perform and using shared learning quickly optimizes those policies. PPO clipping the loss function to limit updates performs better than other on and off-policy algorithms. Traditional MARL algorithms take too long to learn and cannot optimize well on their own for such complex tasks. Additionally, the 95% confidence interval for traditional MARL algorithms is much wider, as the non-determinism in joint observations and drastically varying joint actions lead to drastically different rewards and much different policy learning. Phase one in the two-phase approach drastically

mitigates these issues by pruning the state-space, which leads to a much faster rise in the early stages of learning and overall much higher evaluation returns.

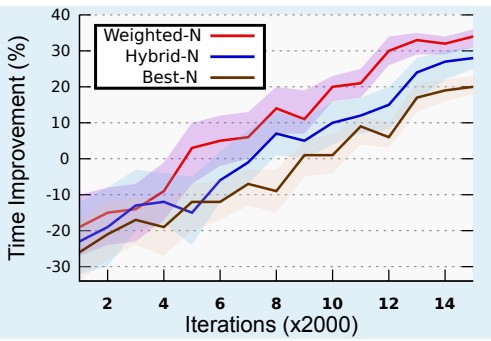

Figure 7: Shared experience learning

Table 1: Trajectory merge summary

| Purpose | Stability KL Divergence | Adaptation KL Divergence | Adaptation Iterations |
|---|---|---|---|
| Weighted-N | 0.0181 | 0.0681 | 8 |
| Hybrid-N | 0.0323 | 0.0776 | 11 |
| Best-N | 0.0206 | 0.0998 | 13 |

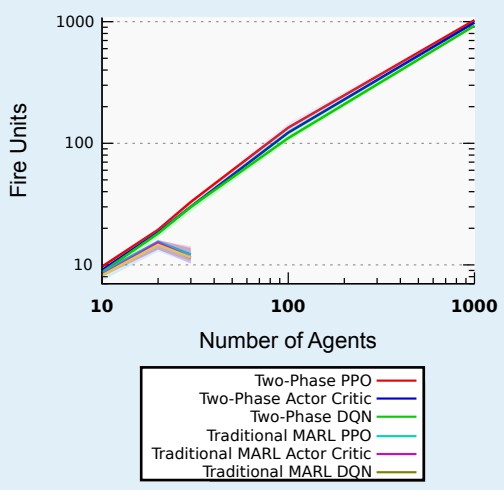

Figure 8: Multi-agent scalability

Figure 8 shows that two-phase algorithms significantly outperform traditional MARL algorithms as the number of agents increases. It was not possible to run tests with more than 30 MARL agents, as the coordination effort significantly increases and the state-space becomes exponential due to many agents. However, the two-phase approach circumvents this issue, allowing for a very large number of agents. This is done by first distributing activities to the best agents capable to handle such activity and then the agents working on those assigned activities, typically either independently or in smaller groups. With groups of homogeneous agents, it becomes possible to use shared experience learning using a population policy bank, making it feasible to learn how to handle very large fires, as shown in the two-phase algorithms in this figure.

Figure 7 shows shared learning performance using three prominent strategies. The Weighted-N strategy performed the best, reaching an average 34% improvement. The ANOVA test yielded a F-statistic of 13.92 and a p-value of $0.0000221 < 0.05$. This indicates a statistically significant difference between the means of the three merge trajectory strategies, contributing to a significant time improvement. Table 1 shows a low KL divergence for the Weighted-N strategy, indicating high stability. Furthermore, with a low adaptation KL divergence, the Weighted-N strategy is resilient to adversarial environment changes, and its low adaptation iteration signifies quick adaptation to diverse new conditions.

These results show that the two-phase multi-agent approach is very effective and scalable in performing large, unpredictable tasks using groups of homogeneous agents.

## 5 CONCLUSION

In this paper, we presented a novel approach to effectively learn how to best perform a dynamic task with multiple groups of homogeneous agents in complex environments. The novel two-phase refocus, refine, repeat approach where phase one evaluates how to best assign the agents to accomplish the task, and phase two refines the performance of the task by using the collective intelligence of the agents to learn an optimal RL policy performs well for such tasks. We demonstrated this approach works quite well with an exemplary system where a large number of drones learn to fight forest fires and tested it using both simulations and with actual drones. This approach can be used in many other applications, including fighting fires in urban settings, providing medical assistance in urban settings, and many disaster relief scenarios.

## REPRODUCIBILITY STATEMENT

This paper contributes an approach for performing complex tasks with a large number of agents. We fully described our proposed approach in section 3 with additional details in A.2 Exemplary Phase Two Algorithms on algorithm implementations. Theorem 1, Prepositions 1 and 2, and Lemma 1.1 give a theoretical basis and are proven in Appendix A.1 Two Phase Approach Proofs. Section 4 and Appendix A.3 Experiment Details give details on obtaining the results. This constitutes complete details on reproducing the work presented in this paper.

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

# A APPENDIX

## A.1 TWO PHASE APPROACH PROOFS

### A.1.1 PROPOSITION 1

**Proposition** (Convergence Acceleration via Merged Learning). If $p$ homogeneous agents merge the top and bottom $n$ % of the combined trajectories, the policy learns $2pn$ times faster than for a single agent learning using all its trajectories.

*Proof.* Since $p$ agents are homogeneous, all trajectories $\mathcal{Z} = \{\zeta_1, \zeta_2, ...\zeta_n\}$ are interchangeable. Therefore, if we were originally getting $s$ trajectories for an agent, we are now getting $sp$ trajectories that are all interchangeable. However, we are only choosing $2n$ % trajectories to train the policy, resulting in $2nsp$ total trajectories. This means if the original training took t timesteps, the new training only takes $t/2pn$ timesteps, which is $2pn$ times faster than a single agent. $\square$

### A.1.2 LEMMA 1.1

**Lemma** (Policy Update through Experience Merging). Updating policy $\pi_{w_{jk}}$ through experience merging with the best and worst $n\%$ trajectories $\zeta$ across all homogeneous agents ensures improvement in expected task performance: $\mathbb{E}[J(W^*(t), X_t^*, \Pi'_{w_{jt}})] \geq \mathbb{E}[J(W^*(t), X_t^*, \Pi_{w_{jt}})]$

*Proof.* The policy gradient theorem states $\nabla_\theta J(\theta) = \mathbb{E}_{\tau \sim p_\theta}[R(\tau) \nabla_\theta \log p_\theta(\tau)]$. Since we select the policies with $n\%$ highest and lowest returns, let there be an indicator function $\mathbf{1}_{sel}(\tau)$, which is 1 if $\tau$ is in the top or bottom $n\%$. Since $E_{\tau \sim p_\theta} \mathbf{1} = 2n$, the expectation of the estimator is: $\nabla_\theta J(\theta) = \mathbb{E}_{\tau \sim p_\theta}\left[\mathbf{1}_{\text{sel}}(\tau) \frac{R(\tau)}{2n} \nabla_\theta \log p_\theta(\tau)\right]$. Since the merge algorithm discards trajectories sets $\mathcal{Z}$ such that $g_t \cdot \nabla_\theta J(\theta_t) < 0$ where $g_t$ is the gradient of a random sample of trajectories from $\mathcal{Z}$, $g_t \cdot \nabla_\theta J(\theta_t) \geq 0$ so the update with best and worst n% of combined trajectories is aligned with $\mathcal{Z}$. With $J \leftarrow J + \alpha \nabla_\theta J(\theta)$, it results in improvement of $J$ by $\alpha \nabla_\theta J(\theta)$. Thus the $2n$ samples result in expected improvement of $\frac{\alpha}{2n} \nabla_\theta J(\theta)$. $\square$

### A.1.3 PROPOSITION 2

**Proposition** (Two-Phase Task Optimization). Let $J(W(t), X_t, \mathcal{B}_g)$ be the task performance function, where $\mathcal{B}_g$ is the policy bank. The iterative execution of phase one and phase two ensures the task policy converges to an optimal solution as the iterations progress if

1. Task decomposition and assignment are comprehensive: $(W^*(t) \text{ and } X_t^*)$, and

2. Policy update through experience merging ensures improvement in expected task performance over time: $\mathbb{E}[J(W^*(t), X_t^*, \Pi'_{w_{jt}})] \geq \mathbb{E}[J(W^*(t), X_t^*, \Pi_{w_{jt}})]$

*Proof.* Since all homogeneous agents $a_i \in \mathcal{A}$ use a particular policy $\pi_{w_{jt}}$, all trajectories $\mathcal{Z} = \{\zeta_1, \zeta_2, ...\zeta_n\}$ are interchangeable and therefore can be treated equivalently. In order to obtain an optimal solution, it is necessary to have a comprehensive task decomposition and assignment result in a policy that improves and convergence, leading to the solution. Additionally, the learning rate $\alpha$ tends to 0 as t tends to infinity. According to Lemma 1.1, if the top and bottom $n\%$ policies from each $\zeta_i \in \mathcal{Z}$ are merged, the $\mathbb{E}(J(W^*(t), X_t^*, \Pi'_{w_{jt}}))$ increases and therefore, the policy improves. Since the two-phase process is continuously repeated, the $\mathbb{E}(J(W^*(t), X_t^*, \Pi'_{w_{jt}}))$ continually converges and improves. $\square$

### A.1.4 THEOREM 1

**Theorem** (Task Learning). If there is a dynamic task $\mathcal{W}(t)$ decomposed and assigned comprehensively as $(W^*(t), X_t^*)$ as described in section 3.1, the task $\mathcal{W}(t)$ can be effectively distributed and learned among agents $a_i \in \mathcal{A}$.

*Proof.* Since the task $\mathcal{W}(t)$ is decomposed and assigned comprehensively i.e. $(W^*(t), X_t^*)$, the constraints $\sum_{i=1}^{N} x_{ij} \geq r(w_{jt})$ and $\sum_{j=1}^{M_t} x_{ij} \leq \kappa(a_i)$ hold true. This ensures that each activity is adequately assigned enough agents and that the agents are not overworked and are capable of working on their assigned activity.

Each activity has an assigned policy from the policy bank, and at least $r(w_{jt})$ agents have trajectories for the activity. By proposition 1, a policy with $r$ agents taking the top and bottom $n$ % of trajectories learns $2rn$ times faster than a single agent since each agent in a group g is homogeneous. So if $r > \frac{1}{2n}$, distributing learning across multiple agents as done in phase two leads to faster learning. Since $r \geq 1$ and $r > \frac{1}{2n}$, by proposition 2 the system continuously learns by merging the policies, allowing for continual evolution and optimal learning. $\qquad\square$

### A.1.5 PRACTICAL CONSIDERATIONS

We believe that our assumptions reflect the practical considerations for the proposed multi-agent reinforcement learning paradigm. The analysis relies on three assumptions, each of which aligns with how large-scale multi-agent systems are actually deployed:

1. Homogeneity within each policy group, assumed by Proposition 1: Policies are used within smaller groups of homogenous agents, grouped by similar sensing, actuation, and capability profile. This is consistent with real-world deployments, where fleets naturally consist of classes of similar types and structures of drones. If a group exhibits internal heterogeneity, it can be further subdivided - our framework imposes no restriction on the number of groups - until this homogeneity is met.

2. Sufficient/comprehensive task decomposition, assumed by Proposition 2 and Theorem 1: This decomposition assumption formalizes the practical goal of full agent utilization, where a task is decomposed into activities such that these activities are assigned to the best capable agent, resulting in the highest possible utilization across the agents. An activity lacking agents with the required capabilities will not be performed effectively, and agents not being assigned sufficient activities leads to underutilization of the available capability.

3. Performance improvement under experience merging, as assumed by Proposition 2: Merging additional trajectories improves the shared policy by incorporating information that helps it adapt. Trajectory sets whose update direction increases expected performance are retained. Thus, every update step moves the policy in an improving direction, ensuring the process is not assumptive but explicitly performance-aligned.

Taken together, these assumptions mirror the operational structure of real multi-agent systems.

## A.2 EXEMPLARY PHASE TWO ALGORITHMS

An *operateAgent* procedure allows a homogeneous agent $a_i \in \mathcal{A}_g$ to execute an activity $\pi_{w_{jt}}$. This procedure is invoked from Algorithm 1. A variety of single-agent on-policy algorithms like ones based on PPO and Actor-Critic and off-policy algorithms like ones based on DQN can be used for the operateAgent procedure that executes activity $\pi_{w_{jt}}$. A sample algorithm based on PPO (Schulman et al., 2017) is shown by algorithm 3

For the forest fire fighting with drones application, the state is the current representation of the drone and its relation with the fire, including location, fire intensity, wind speed, wind direction, humidity, distance to nearest settlement, distance to body of water, etc. The action is the discrete actions the drones can do, including moving in a certain direction, spraying water in a certain direction or intensity, or creating a controlled fire. Since the number of states and actions is reasonable for each particular policy due to the distributed approach, it is able to learn it in a reasonable timeframe.

---

**Algorithm 3** operateAgent: Proximal Policy Optimization (PPO)

---

**Require:** Agent identifier $a_i$, starting policy $\Pi_i^k(b)$
**Ensure:** Returns a set of trajectories $(h_i^t, a_i^t, r_i^t, h_i^{t+1})$
 1: Initialize actor network $\pi$ with parameters $\phi$
 2: Initialize critic network $V$ with parameters $\theta$
 3: Initialize policy $\pi \leftarrow \Pi_i^k(b)$
 4: Initialize empty trajectories set $\mathcal{D}_i$
 5: **for** each episode **do**
 6:     **for** time step $t = 0, 1, 2, \ldots$ **do**
 7:         Observe current state $h_i^t$
 8:         Sample action $a_i^t \sim \pi(\cdot | h_i^t; \phi)$
 9:         Apply action $a_i^t$; observe reward $r_i^t$ and next state $h_i^{t+1}$
 10:        $D_i = D_i \cup \langle h_i^t, a_i^t, r_i^t, h_i^{t+1} \rangle$
 11:        $\pi_\beta(a_i^t | h_i^t) \leftarrow \pi(a_i^t | h_i^t; \phi)$
 12:        **for** epoch $e = 1, \ldots, N_e$ **do**
 13:           Importance sampling ratio: $\rho(h_i^t, a_i^t) \leftarrow \frac{\pi(a_i^t | h_i^t; \phi)}{\pi_\beta(a_i^t | h_i^t)}$
 14:           $N$-step: $Adv(h_i^t, a_i^t) = \sum_{\tau=0}^{N-1} \gamma^\tau R(h_i^{t+\tau}, a_i^{t+\tau}, h_i^{t+\tau+1}) + \gamma^N V(h_i^{t+N}) - V(h_i^t)$
 15:           Target: $y_i^t \leftarrow \sum_{\tau=0}^{N-1} \gamma^\tau r_i^{t+\tau} + \gamma^N V(h_i^{t+N})$
 16:           Entropy regularization: $\mathcal{H}(\pi(\cdot | h_i^t; \phi)) = \sum_{a \in \mathcal{A}} \pi(a | h_i^t; \phi) \log \pi(a | h_i^t; \phi)$
 17:           Actor loss: $\mathcal{L}(\phi) \leftarrow -\min \Big[ \rho(h_i^t, a_i^t) \cdot Adv(h_i^t, a_i^t),$

$$\text{clip}(\rho(h_i^t, a_i^t), 1 - \epsilon, 1 + \epsilon)$$
$$\cdot Adv(h_i^t, a_i^t) \Big] - \alpha \mathcal{H}(\pi(\cdot | h_i^t; \phi))$$

 18:           Critic loss: $\mathcal{L}(\theta) \leftarrow (y_i^t - V(h_i^t; \theta))^2$
 19:           Update parameters $\phi$ by minimizing actor loss $\mathcal{L}(\phi)$
 20:           Update parametetrajectoryTablers $\theta$ by minimizing critic loss $\mathcal{L}(\theta)$
 21:        **end for**
 22:     **end for**
 23: **end for**
 24: **return** $\mathcal{D}_i$

---

A set of trajectories $\zeta$ is selected across the group of agents $\mathcal{A}_g$ to merge shared experiences back to the policy $\pi_{w_{jt}}$ before placing it back in the policy bank. The *updateSharedLearning* procedure is invoked by Algorithm 2 to merge shared learning across the agents. A variety of single-agent on-policy algorithms like ones based on PPO and Actor-Critic and off-policy algorithms like ones based on DQN can be used for *updateSharedLearning* procedure alongside the corresponding *operateAgent* procedure. Here, we illustrate an *updateSharedLearning* algorithm based on actor-critic (Konda & Tsitsiklis, 2000) to merge a selection of trajectories $\mathcal{Z} = \{\zeta_1, \cdots, \zeta_n\}$ obtained from homogeneous agents $a_i \in \mathcal{A}_g$ while they executed activity $w_{jt}$ using policy $\pi_{w_{jt}}$.

---

**Algorithm 4** updateSharedLearning: On-Policy Experience Sharing

---

**Require:** Shared policy $\Pi^k(b)$, shared experience buffer $\{D_{shared}\}$
**Ensure:** Updated shared policy $\Pi^k(b)$ with off-policy corrections
 1: Initialize temporary policy $\Pi^k_{temp}(b) \leftarrow \Pi^k(b)$
 2: **for** each epoch $e = 1, \ldots, N_e$ **do**
 3:     **for** each mini-batch of transitions $(h^k, a^k, r^k, h^{k+1})$ sampled from $D_{shared}$ **do**
 4:         **for** each agent $i$ **do**
 5:             Importance sampling ratio correcting off-policy updates: $\rho(h_i^k, a_i^k) \leftarrow \frac{\pi(a_i^k|h_i^k;\phi_i)}{\pi_\beta(a_i^k|h_i^k)}$
 6:             N-step Adv: $Adv(h_i^k, a_i^k) = \sum_{\tau=0}^{N-1} \gamma^\tau R(h_i^{k+\tau}, a_i^{k+\tau}, h_i^{k+\tau+1}) + \gamma^N V(h_i^{k+N};\theta_i) - V(h_i^k;\theta_i)$
 7:             Target: $y_i^k \leftarrow r_i^k + \gamma \max_{a_i' \in A_i} Q(h_i^{k+1}, a_i'; \bar{\theta}_i)$
 8:             Corrected actor loss: $\mathcal{L}(\phi_i) = -\rho(h_i^k, a_i^k) \left( r^k + \gamma V(h^{k+1};\theta_i) - V(h^k;\theta_i) \right) \log \pi(a_i^k|h_i^k;\phi_i)$
 9:             Critic loss: $\mathcal{L}(\theta_i) \leftarrow \frac{1}{B} \sum_{k=1}^{B} (y_i^k - Q(h_i^k, a_i^k;\theta_i))^2$
10:             Update actor parameters $\phi_i$ by minimizing $\mathcal{L}(\phi_i)$
11:             Update critic parameters $\theta_i$ by minimizing $\mathcal{L}(\theta_i)$
12:         **end for**
13:     **end for**
14: **end for**
15: Update shared policy $\Pi^k(b) \leftarrow \Pi^k_{temp}(b)$ using aggregated policy updates
16: **return** Updated shared policy $\Pi^k(b)$

---

## A.3 EXPERIMENT DETAILS

### A.3.1 SETUP AND PARAMETERS

Various aspects of the two-phase approach were tested with experiments using an exemplary forest-fighting system disclosed in section 4.1. A simulated agent was operated using a set of test fire-images and corresponding sensor data for that image. Inputs from many agents are reported to a task distributor that performs the task distribution. A real agent is an actual fire-fighting drone that captures the fire-image using its camera and acquires current sensor data using its on-board sensors to correspond with the captured image. This data is periodically sent to a task distributor to reassign activities to each agent. The test fire-images were input to the image pipeline, and the sensor data were input to the sensor pipeline as shown in Figure 2a. The task distribution result assigns a hotspot or an edge to an agent. Such an assignment is reported to the agent as an activity assignment. An agent continuously performs its assigned activity as shown in the Figure 2b. A reassignment of a different activity by the task distributor results in the agent preempting its current assigned activity and moving to the new assigned activity. Best and worst trajectories across similar agents performing an activity are used for merging their shared experiences into their shared policy persisted in the policy bank.

A real agent is a Raspberry-Pi-based drone exemplary agent that is an X-Configuration Quadcopter UAV with a PixHawk 2.4.8 flight controller driving A2212/KV930 motors with 8038 propellers and SimonK 10A ESC, a GPS M8N, and a Matek Optical Flow sensor for positioning, along with Benewake TFmini Plus LIDAR sensor. Drone captures temperature, humidity, pressure, wind speed, and wind direction using onboard sensors along with image frames using Raspberry Pi Camera, and reports them for task distribution by default every 15 seconds. Image resolution defaults to 384 x 384. It communicates directly with a ground control station using onboard WiFi, and falls back to radio telemetry if WiFi is out of range.

Simulated agents are pure software components that run on multiple servers with 32 cores, 128GB RAM, and 1TB storage medium end servers. These agents use a pre-captured stream of image and sensor data with around 4800 samples for different fire scenarios. The task distribution uses multiple 16GB vRAM GPUs, depending on the number of agents and fire analysis request frequency for each experiment.

Different fire scenarios are simulated based on the actual fire dataset. In a virtual agent, the duration and frequency of spray operation are recorded to determine the effect of the fire extinguisher in changing the fire based on modeling guidelines in Hansen (2012). This change helps in computing the reward for the current action of the agent. When using a real agent, it is also necessary to precisely recreate a test environment so that the performance of a real agent can be evaluated in conditions close to those of a real forest fire. Based on an actual fire sample from the fire dataset, a forest fire mock layout is created on the ground using different fire-colored fabric pieces as shown in 10. The fabric is moved to simulate changes in the fire condition. Images captured by a real agent is processed using a CNN model that is trained on these forest fire mock layouts. A point-laser device operated by the real agent is used to simulate the spraying of a fire extinguisher. The duration and frequency of this laser operation are recorded, and using modeling guidelines in Hansen (2012), the effect of the fire extinguisher is determined to guide altering the fire status on the ground. In order to streamline results with real and virtual agents, a fire-unit is used to represent one unit of fire. By default, one fire-unit maps to one square kilometer of a real forest fire, and this is typically equivalent to one square centimeter of the forest fire mock layout. Fire unit is used to represent the size of fire for all results in section 4.2.

### A.3.2 AGENTS, ACTIVITY ASSIGNMENTS, AND EXECUTION FOR EXEMPLARY SYSTEM

The formal task decomposition and activity distribution is disclosed in section 3.1. Here, we explore certain aspects of this formalism in a more informal setting as applied to the exemplary system of Section 3.3 for providing a deeper understanding of the underlying concepts.

The forest fire-fighting task $\mathcal{W}(t)$ changes over time as fire spreads or is contained. Agents are systems with specific capabilities that help perform activities related to the task of fighting wildfires. This may include drones of different sizes, speeds, and their ability to perform the firefighting tasks. Agents are categorized into groups based on their capabilities, which for this exemplary domain includes fire-extinguishing capacity, fire-extinguishing type, and drone speed. All agents in the same group have the same capabilities. E.g., we have a group of small, medium, and large drones with relative speeds 4x, 2x, 1x, and liquid fire extinguisher capabilities of 10 liters, 50 liters, and 100 liters, respectively. A drone may temporarily leave its group, such as to refuel and join back when it is ready to operate again. However, a drone does not change groups, as the drone's association with a group is based on its capabilities.

During phase-1, the current task is holistically analyzed and decomposed into many activities, such as fighting a specific fire edge or a specific fire hotspot at a specific location in the forest. Each such activity $w_{jt}$ involves a complexity level $c(w_{jt})$, such as the danger it poses and the likelihood of it spreading the fire. An activity of a specific complexity level needs to be addressed by agents with a specific capability. E.g., a fire edge near an inert area like a lake or a rocky hill is not very dangerous and may be handled by a small, low-capacity drone, whereas one that is close to dangerous vegetation requires a medium or high-capacity drone that can immediately contain it. An activity such as fighting fire-edge or fighting a hotspot is assigned to each agent, not their group. An activity may require one or more agents as defined by $r(w_{jt})$. E.g., when $r(w_{jt}) = 2$, two or more agents, possibly from different agent groups, may be assigned to $w_{jt}$. Thus, two groups may have agents working on an overlapping subset of activities. Moreover, $\kappa(a_i)$ is the maximum number of activities that agent $a_i$ can handle at a given time. The $r(w_{jt})$ and $\kappa(a_i)$ are typically prior domain knowledge specified by the expert or pre-defined while creating the domain. This allows defining more than one drone to be assigned to an activity and more than one activity assigned to a drone for maximum flexibility. A very big hotspot cannot be handled by a single drone - requiring many drones, and a large drone may handle many small fire edges.

An agent is not required to operate on its assigned activity to completion. An agent's task assignment may be continuously revised, and the agent may not be tied to an activity until completion. We iterate continuously between the two phases. In phase-1, a holistic view of the current task $\mathcal{W}(t)$ governs the partitioning of the task into activities, and assignment of agents to these activities. An agent continues to work on an activity until the activity is completed or the agent gets reassigned. E.g., a small drone may be assigned to a fire edge that was initially of low risk. However, due to a change in wind direction, that fire edge is now flagged as high risk, causing it to be assigned to another potent medium or high capacity drone. Upon completion of any activity, it can no longer be assigned to any agent.

If the task is decomposed such that each agent is assigned the most appropriate activity, it can lead to optimal results. The idea is for decomposition to shoot for full agent utilization so that all available agents remain active. So $\mathcal{D}$ partitions the task, assigning activities to agents based on their capability. With excess activities, some activities won't get done immediately. If there are excess agents, some agents remain idle. We try to avoid this by attempting $|\mathcal{W}_g(t)| \geq |\mathcal{A}_g|$.

The custom test environment was specifically designed to evaluate scalability issues with many agents. Same codebase between virtual and real drones enables them to coexist with real drones using camera and sensors against sampling of these inputs from fire datasets, along with simulated activities for virtual drones. Real drones operate alongside virtual or other real peers, all coexisting under a common custom ground control implementation that launches virtual drones with special virtual settings. This design allows us to test performance in the presence of a very large number of agents with real messages, unearthing any scalability issues and communication delays that would be encountered with a very large number of real drones, using the test setup with many virtual drones. The purpose of such a hybrid setup was also to visualize how real drones perform their activities in the presence of a very large number of other real/virtual drones. The testing environment can simulate more than 3000 drones, and two-phase algorithm testing shows effective scaling beyond 1000 agents. To compare two-phase algorithms against SOTA MARL algorithms, we had to limit test comparisons to only 25 agents, as SOTA MARL algorithms failed beyond 25 agents.

Note that the scope of this paper is a novel approach to enable groups of homogeneous agents to autonomously learn to perform unpredictable tasks, including those with a massive state space not feasible with today's state-of-the-art approaches. We use the exemplary firefighting domain to demonstrate various aspects of our novel approach, and the real drones, virtual drones, and associated controls constitute an effective testbed for testing these aspects.

### A.3.3 CONTAINMENT PERFORMANCE STUDY

This experiment evaluated the performance of the two-phase approach against a baseline of actual fire fighters. It evaluates the improvement of the containment time and the fire extinguisher resources needed to reach that containment against the baseline. For a specific target fire sample scenario, using the information from the datasets, we obtained additional information related to the fire such as vegetation, containment time, and percentages. This information was then correlated with the model to obtain the fire containment time and resources involved in fighting the fire. Based on the fire size, groups of homogeneous agents are used with a fleet comprising 50% small capacity drones, 35% medium capacity drones, and 15% large capacity drones. The drones used a pre-trained population policy bank. The containment time included the time the drones are armed to the time the entire fire is extinguished. Moreover, each drone recorded the total amount of fire extinguisher used, and these were compared against the baseline of real fire fighters. The test was repeated for fires of different sizes and hotspots. The same test was repeated for multiple trials on samples with 10 and 30 hotspots.

### A.3.4 ABLATION STUDY

The task distribution is performed during phase one processing, and it can have a profound impact on the overall performance. Since there are multiple components for performing this task distribution, an ablation study was performed to determine the necessary components for optimal task distribution. The transformer was replaced by image-analysis-based hotspot detector, the A* component was replaced by a rule-based task assigner, and the edge-progression component was replaced by a contour-based edge processing. A component was swapped out, and the evaluation return was recorded to identify which components provide optimal performance.

### A.3.5 TWO-PHASE ALGORITHMS STUDY

Upon assignment of an activity, each agent loads a policy from the population policy bank and performs activity steps under the guidance of this policy. The efficacy of this algorithm directly impacts the efficacy of the overall approach, and therefore, different algorithms are evaluated to determine which algorithms provide optimal performance. The policies are not pre-trained - the test uses the evaluation returns as agents learn policies and execute activity steps using these policies. On-policy PPO was evaluated with a clipping epsilon of 0.1 with a policy gradient actor and critic models with two layers of 128 nodes.

The primary purpose of experimentation was to evaluate the two-phase approach using an exemplary fire-fighting domain, testing key aspects of our approach. Each algorithm in Figure 6 was evaluated with the optimal set of hyperparameters obtained after testing for these cases. For two-phase PPO MARL, an $\epsilon$ clip value of 0.1 and a continuously decreasing learning rate provided optimal performance. Larger PPO clipping thresholds (0.2–0.3) produced overly aggressive updates and led to moderate reductions in evaluation returns for Two-Phase MAPPO. Higher initial learning rates caused similar degradation in both two-phase PPO and two-phase A2C, reflecting their reliance on stable value estimates. Adjusting the discount factor away from 0.99 also impacted performance, affecting PPO, A2C, and DQN to varying degrees. Reward parameters of $\alpha$ and $\beta$ represented as 2500 and 3500. While observing the difference in reward would be largely ineffective since higher values would implicitly result in a higher value of evaluation returns, these values enabled prioritizing greater emphasis on area reduction over intensity reduction, resulting in greater refinement efficacy.

The actor-critic policy also used models with two layers of 128 nodes, and DQN used a Q and target networks with two layers of 128 nodes. A shared experience with Weighted-N trajectory merging strategy was used to merge the experiences of homogeneous agents sharing similar activities. Traditional MARL Algorithms tested include Centralized Training Decentralized Execution Actor-Critic, PPO, and DQN Algorithms. Since traditional MARL algorithms do not perform well, this testing was done using 25 agents to compare the efficacy of the two-phase algorithm versus traditional MARL algorithms. The tests were performed for different environment timesteps ranging from 2000 to 20000 time steps.

### A.3.6    MULTIAGENT SCALABILITY STUDY

The two-phase algorithms study was further extended to evaluate performance with a different number of agents. Each agent was assigned a hotspot spanning a fire-unit and allowed to perform the activity for a total of 2000 timesteps. Upon completion, the amount of fire extinguished across all agents is computed to determine the effective total number of fire-units that were collectively extinguished across these agents. The number 2000 timesteps was chosen to allow an agent sufficient time to extinguish a large portion of the fire. It must be noted that since traditional MARL does not scale well beyond around 30 agents, the tests were conducted with only two-phase algorithms beyond 30 agents.

For the trajectory merge test as in Table 1, the KL divergence shows the difference in the probability distributions. For this paper, it is used to show the improvement of policy refinement through trajectory merging. Stability KL Divergence is the difference in the distributions between the current stable distribution and minor perturbations affecting that stability. Adaptation KL Divergence is the difference in the distributions between the original distribution and a restabilized distribution that has undergone major perturbations such as drastic changes in wind speeds and humidity. Adaptation Iterations is the number of phase-one -> phase-two cycle iterations that it takes to reach the accuracy of the current domain, to see how quickly the system can adapt to different environments.

### A.3.7    SHARED EXPERIENCE LEARNING STUDY

This experiment was conducted to study the efficacy of merged experience learning using trajectories from homogeneous agents with similar capabilities performing a similar activity. Because each group of agents maintains its own specialized policy and only merges experiences internally, we do not observe any policy destabilization. Unlike the conventional population-based training for policy space response oracle (PSRO) as in Lanctot et al. (2017) for non-cooperative tasks, here, the cooperating agents learn by sharing their experiences upon completion of an activity and the goal is to determine an optimal way to merge the experiences captured in the trajectory of these agents. Trajectories are compared based on a reward for a step in the trajectory. The best-N strategy was tested by selecting only N-best trajectories from the reporting agents, N typically set to one-fourth of total homogeneous agents reporting their trajectories. However, worst experiences also teach what not to do and therefore, a hybrid-N strategy was also tested with best-N and worst-N trajectories. Another variant of the hybrid strategy is the Weighted-N strategy, where the best and worst strategies are given the highest weight among the best-N and worst-N trajectories. More weight causes a trajectory to be repeatedly used that many times for experience learning, and each of the N best and worst trajectories is weighted based on their ranking. A policy gets saved in the population policy bank upon shared learning, and this policy gets distributed across agents, serving as a critical means to communicate

and share experiences across the agents. Therefore, the efficacy of shared experience learning forms an important aspect of the two-phase learning approach.

## A.4 INJECTING EXTERNAL/DOMAIN KNOWLEDGE

A unique aspect of this approach is the ability to complement pure reinforcement learning with adjunct strategies, including domain intelligence, learning algorithms, human-in-the-loop (HIL), to expedite learning. It provides a means to inject prevalent external/domain knowledge in the learning process, making it feasible to significantly prune the massive search space. Figure 8 shows how the current state-of-the-art MARL approaches can't scale for such a massive state space, limiting their real-world MARL applicability. Most MARL approaches fail to inject a means to curb exploring irrelevant portions of massive search-space, resulting in their failure for pragmatic real-world use of such complex huge tasks.

Injecting prevalent external/domain knowledge using a phase-1 strategy enables significantly pruning search space resulting in phase-2 learning for huge, complex tasks which are not possible today. Phase-2 "refine" is completely task-independent, and it is also possible to transfer optimizations similar to Phase-1 optimizations using sensory and image data demonstrated in the exemplary system to other domains. E.g., Locating fire-areas using transformer pipelines can be transferred to locating flooded areas for a system of autonomous robots in a flood-control application. Fire-fighting activities of exemplary system are replaced by flood-control activities that robots learn in identical manner for the flood-control application. Thus a system similar to the exemplary system disclosed here for fighting forest fires can be used to model many other applications that tackle complex tasks in dynamic and unpredictable environments.

## A.5 TWO PHASE IDEATION WITH SMAC V2

### A.5.1 IDEATION STRATEGY

In the early stages of our research, we developed our ideation using SMACv2 as in Ellis et al. (2023) to experiment with how to prune a large RL search space. SMACv2 provides a standard way to compare performance against many state-of-the-art algorithms in small-to-moderate environments, and unlike its predecessor, SMAC as in Samvelyan et al. (2019), it provides for a larger RL search space and partial observability to experiment with diverse scenarios for a small number of agents. In our explorations with both SMACv2 and SMAC, we quickly faced severe scaling issues with the state-of-the-art algorithms as well as the test frameworks as we tried to increase the number of agents. So we had to limit our explorations with a small number of agents supported by the test framework and the state-of-the-art algorithms and use the exemplary forest firefighting environment in Section 4 for comprehensive testing of all aspects of our research. Although the SMACv2 test framework and baseline state-of-the-art algorithms were limited and the behaviors vary significantly, it nevertheless allowed us to quickly experiment with different strategies in the early stage of our research that led to our two-phase approach and compare them with state-of-the-art algorithms, revealing many interesting insights which we share here.

SMACv2 procedurally generates teams for different races - Terran, Zerg, and Protoss. Terran uses ranged attacking units of Marine and Marauder, as well as Medivac support units. Zerg uses a mixture of ranged unit Hydralisk, melee unit Zergling, and exploder unit Baneling. For our tests, we used Terran and Zerg, as Terran allows testing range-focused strategy and Zerg enables testing a hybrid strategy.

To explore ideations for the two-phase approach, we used a phase-1 strategy that executes part of a predefined combat strategy suitable for fighting the enemy units, and phase-2 involved using reinforcement learning to learn the remainder of the combat strategy. The phase-1 strategy allows the agent to prune out the RL search space by eliminating moves that do not conform to the predefined combat strategy, allowing phase-2 to then learn for a smaller RL search space. For example, a combat strategy involves positioning the units relative to allies and enemy units and attacking the enemy with the right weapons and timing. Learning both positioning and firing involves a huge RL search space with many units. As the phase-1 strategy guides the agent to the correct position, the agent then has to learn firing-related behavior, significantly reducing the RL search space. The tables show the

improvement of the two-phase strategy over a baseline state-of-the-art algorithm with both average improvement and range of improvements observed during multiple trials.

### A.5.2 EFFECTIVE TERRAN COMBAT

Terran's ranged composition imposes some unique coordination requirements. Marauders are durable armored frontliners that intercept and deliver high single-target damage, slowing enemies to control the pace of engagement. While they have ranged attacking capabilities, their range is shorter than the Marines. Marines are more vulnerable but can provide longer-range, effective bursts of sustained DPS - so Marines must hide behind Marauders. Medivacs heal damaged units but cannot fight back - so they must remain sealed behind other allies. An effective strategy, therefore, coordinates Marauder positioning, Marine focus fire and kiting, and Medivac healing, all operating as a cohesive group for maximum efficacy.

### A.5.3 RANGED STRATEGY

For Terran scenarios (Terran_5_vs_5 and Terran_20_vs_20), we employ a structured ranged-unit strategy that separates positioning and firing into two phases to evaluate our two-phase approach. Phase-1 attempts the predefined spatial formation with Marauders taking positions in the front facing the enemy, Marines aligned behind them, and Medivacs maintaining a protected rear position, while enforcing sufficient spacing, enemy-facing orientation, and engaging in limited micro-adjustments. By eliminating random formation-breaking movements, Phase-1 dramatically narrows the effective RL search space, converting chaotic navigation into focused positional behaviors. With a focused positioning, Phase-2 learns firing-related decisions for combat effectiveness, including focus-fire selection, target switching, kiting, burst timing, and Medivac healing prioritization.

As shown in Table 2, this strategy achieved faster and higher battle win rates than the baseline QMIX algorithm. Our results showed a steep early rise in performance compared to baseline, confirming that Phase-1's structured positioning, by replacing random positioning movement with strategy-focused aligned movements, sharply narrows the effective exploration space and enables faster learning. With agents consistently placed in tactically favorable formations, Phase-2 can immediately begin learning coordinated firing behaviors rather than spending millions of steps discovering viable positions. In contrast, QMIX requires significantly longer training to reach moderate win percentages and also fails to match the peak performance achieved by our method. The sustained advantage over 10M timesteps highlights that disciplined, strategy-aligned positioning not only accelerates convergence but also enables higher-quality policies in larger range combat scenarios. As evident with Terran_20_vs_20, as the search space increases, its impact becomes even more significant. Similar results were obtained against baseline MAPPO as shown in Table 3. Use of different Phase-2 algorithms did not significantly alter the results. Phase-1 preserves the essential tactical decisions but removes the combinatorial explosion associated with free movement, enabling significantly faster and more stable Phase-2 learning, resulting in better overall efficacy across both small and large Terran engagements.

Table 2: Terran 2-Phase over QMIX

| Time steps ($10^6$) | Terran_5_vs_5 Improvement % | | | Terran_20_vs_20 Improvement % | | |
|---|---|---|---|---|---|---|
| | Avg. | Max. | Min. | Avg. | Max. | Min. |
| 1 | 16.2 | 28.1 | 9.6 | 28.7 | 35.6 | 16.5 |
| 2 | 8.4 | 19.3 | -3.2 | 20.3 | 35.1 | 4.7 |
| 3 | 2.8 | 13.3 | -8.5 | 9.5 | 16.8 | -2.3 |
| 4 | 5.3 | 17.4 | -8.1 | 5.3 | 16.0 | -5.8 |
| 5 | 2.7 | 11.6 | -7.3 | 9.1 | 21.8 | 0.7 |
| 6 | 8.1 | 19.2 | 0.8 | 7.9 | 17.3 | 0.3 |
| 7 | 3.8 | 14.6 | -3.5 | 7.2 | 19.7 | -1.4 |
| 8 | 6.2 | 12.7 | -1.9 | 8.8 | 22.6 | -5.0 |
| 9 | 4.8 | 13.3 | -5.3 | 6.4 | 18.2 | -1.9 |
| 10 | 4.0 | 17.4 | -6.3 | 7.1 | 21.4 | -2.6 |

Table 3: Terran 2-Phase over MAPPO

| Time steps ($10^6$) | Terran_5_vs_5 Improvement % | | | Terran_20_vs_20 Improvement % | | |
|---|---|---|---|---|---|---|
| | Avg. | Max. | Min. | Avg. | Max. | Min. |
| 1 | 40.6 | 52.3 | 30.7 | 38.3 | 44.1 | 26.2 |
| 2 | 31.3 | 42.5 | 20.7 | 32.8 | 43.3 | 19.2 |
| 3 | 22.2 | 33.6 | 12.8 | 32.0 | 40.6 | 22.9 |
| 4 | 25.5 | 38.4 | 10.9 | 28.5 | 36.2 | 15.7 |
| 5 | 23.8 | 34.2 | 12.3 | 25.4 | 37.2 | 16.5 |
| 6 | 27.1 | 41.0 | 15.7 | 22.2 | 29.8 | 12.3 |
| 7 | 17.8 | 24.7 | 11.1 | 17.7 | 25.3 | 8.9 |
| 8 | 19.5 | 32.2 | 5.6 | 18.7 | 27.2 | 7.5 |
| 9 | 21.2 | 30.6 | 7.9 | 16.3 | 25.0 | 3.8 |
| 10 | 20.7 | 31.2 | 10.4 | 15.6 | 26.8 | 4.6 |

### A.5.4 EFFECTIVE ZERG COMBAT

Zerg's hybrid composition imposes unique coordination requirements that are different than those of Terran. Zerglings must reach and touch enemy units so that higher-value enemy Banelings and fragile backline enemy units become accessible. Banelings are scarce, high-impact resources whose explosions provide greater impact when targeting dense enemy clusters rather than isolated dying units. Hydralisks are ranged units providing sustained DPS with clear firing lanes, and remain protected when behind the melee screen. An effective strategy must therefore synchronize Zergling engagement, Baneling explosion timing, and Hydralisk focus fire into a coherent, staged attack.

### A.5.5 HYBRID STRATEGY

For Zerg scenarios (Zerg_5_vs_5 and Zerg_20_vs_20), the ranged strategy used for Terran is insufficient, as unit roles differ, calling for a different hybrid combat strategy. For the hybrid strategy, Phase-1 arranges units into multiple spatial group configurations, each group comprising a small number of Zerglings forming a melee screen in front, one or two Banelings immediately behind the Zerglings, and a few Hydralisk in the rear. These groups are placed side by side with enough spacing between groups such that an enemy Baneling explosion damages only a single group, while neighboring groups continue their fight. This structured positioning strategy purges many chaotic movement patterns so that Phase-2 can focus on learning Baneling commit timing, local surroundings, and Hydralisk target selection and firing patterns for effective hybrid Zerg combat.

As shown in Table 4, with this hybrid strategy, learning was much earlier than baseline QMIX, as there is less to discover initially in terms of viable movement, resulting in a high latency before QMIX becomes useful. QMIX fails to precisely master all nuances of positioning and firing, and its performance remains below our two-phase strategy even after 10M+ timesteps. In contrast, Phase-2 successfully masters detailed firing patterns and their coordination with the Phase-1 movements, yielding highly effective hybrid Zerg behavior. QMIX struggles substantially on the more challenging Zerg_20_vs_20 scenario: its win rate increases slowly and remains well below our method, reflecting the difficulty of exploring an enormous joint movement and firing space without structural guidance. In contrast, our two-phase strategy performs consistently well. Similar results were obtained against baseline MAPPO as shown in Table 5. These results reveal that aggressively reducing the effective RL search space and guiding exploration based on a good combat strategy is particularly beneficial when the underlying search space is very large.

This exploration led to some very interesting insights that helped the ideation of our two-phase approach. When the RL search space is unreasonably large, the SOTA algorithms fail to adequately learn in a reasonable time and hence are of little pragmatic use. The problem becomes worse as the problem space becomes bigger. Injecting a strategy that continuously targets trimming the search space while working alongside the learning algorithm significantly expedites learning and leads to effective learning for these problems.

Table 4: Zerg 2-Phase over QMIX

| Time steps (10^6) | Zerg_5_vs_5 Improvement % | | | Zerg_20_vs_20 Improvement % | | |
|---|---|---|---|---|---|---|
| | Avg. | Max. | Min. | Avg. | Max. | Min. |
| 1 | 12.7 | 25.3 | -4.2 | 11.9 | 19.8 | 1.1 |
| 2 | 11.6 | 20.8 | -2.5 | 9.7 | 22.1 | -8.4 |
| 3 | 11.2 | 18.3 | -7.0 | 8.4 | 21.6 | -8.7 |
| 4 | 9.4 | 19.9 | -2.8 | 6.7 | 18.3 | -7.5 |
| 5 | 3.7 | 19.4 | -5.5 | 7.0 | 19.4 | -8.8 |
| 6 | 8.2 | 21.7 | -1.9 | 9.3 | 21.7 | -4.9 |
| 7 | 9.2 | 23.3 | -4.1 | 11.3 | 22.2 | -3.6 |
| 8 | 10.0 | 26.6 | -2.3 | 12.6 | 22.6 | -3.0 |
| 9 | 9.5 | 19.8 | -2.7 | 14.7 | 28.3 | 0.3 |
| 10 | 10.8 | 20.1 | -4.8 | 13.9 | 26.2 | -2.1 |

Table 5: Zerg 2-Phase over MAPPO

| Time steps (10^6) | Zerg_5_vs_5 Improvement % | | | Zerg_20_vs_20 Improvement % | | |
|---|---|---|---|---|---|---|
| | Avg. | Max. | Min. | Avg. | Max. | Min. |
| 1 | 26.3 | 44.2 | 9.8 | 26.4 | 31.1 | 17.0 |
| 2 | 29.5 | 41.6 | 14.3 | 33.1 | 41.3 | 22.7 |
| 3 | 22.7 | 36.3 | 4.7 | 28.9 | 40.6 | 14.6 |
| 4 | 18.8 | 31.2 | 5.0 | 22.4 | 34.7 | 11.1 |
| 5 | 7.3 | 22.6 | -5.7 | 19.5 | 31.4 | 4.6 |
| 6 | 12.7 | 23.0 | -1.9 | 13.6 | 28.2 | 2.1 |
| 7 | 10.6 | 24.2 | -5.3 | 13.0 | 30.3 | -3.8 |
| 8 | 9.2 | 24.4 | -4.3 | 12.2 | 29.7 | -5.2 |
| 9 | 9.7 | 21.6 | -3.9 | 13.5 | 25.8 | 0.3 |
| 10 | 8.8 | 22.4 | -5.3 | 8.6 | 25.1 | -6.5 |

## A.6 FIRE-FIGHTING WITH DRONES

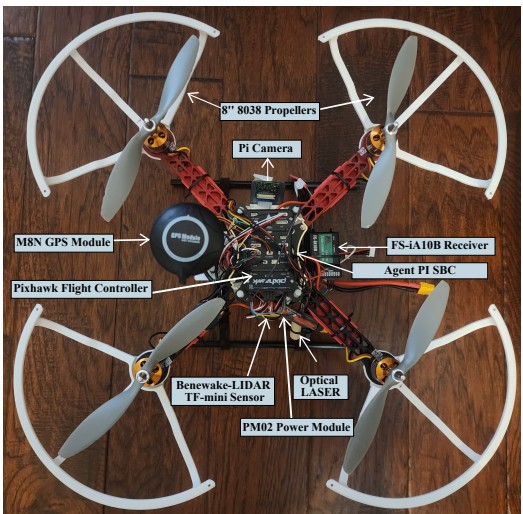

Figure 9: Drone top view

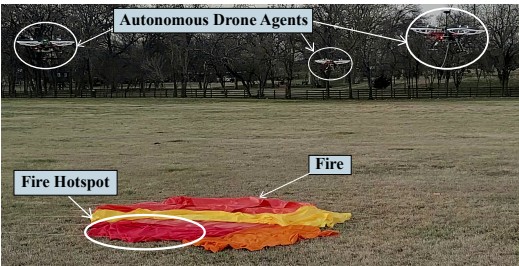

Figure 10: Drones in action

