# OpenReview forum: "Scalable Multi-Agent Autonomous Learning in Complex Unpredictable Environments"
_ICLR.cc/2026/Conference — Submitted to ICLR 2026_

### Official Review · Reviewer_ZeLm · 2025-10-25

**Soundness:** 3
**Presentation:** 3
**Contribution:** 3
**Rating:** 6
**Confidence:** 3

**Summary:**

This paper proposes a novel two-phase iterative multi-agent reinforcement learning (MARL) framework for scalable autonomous learning in large, dynamic, and unpredictable environments. The approach targets cooperative tasks involving groups of homogeneous agents, such as drones in forest firefighting. In Phase 1 (Refocus), a task distributor decomposes the overall task into activities based on the current environment state (e.g., using heuristics like A* planning, combined with domain-specific pipelines for fire edge progression and hotspot prediction via transformers) and assigns them to suitable agents, reducing the state space and coordination overhead. In Phase 2 (Refine), agents execute their assigned activities using policies from a shared policy bank, collect trajectories, and refine policies through merged shared experiences (e.g., selecting and merging best/worst trajectories from homogeneous agents). The framework iterates between phases for continuous adaptation, leveraging algorithms like PPO, Actor-Critic, or DQN for policy updates.

**Strengths:**

1. The framework addresses a critical gap in MARL: handling thousands of agents in non-stationary, partial-observable environments. By decoupling task distribution (Phase 1) from policy refinement (Phase 2), it avoids the exponential state-action space explosion in joint MARL. The use of homogeneous agent groups and shared policy banks enables efficient experience sharing, leading to faster learning
2. Experiments cover ablation studies (e.g., removing transformers reduces returns by ~50%), comparisons to MARL baselines (two-phase PPO outperforms MAPPO by 2-3x in returns), scalability tests (up to 1,000 agents vs. MARL's limit of ~30), and statistical validation (ANOVA p<0.05 for trajectory strategies). Results on real fires (e.g., 40%+ time savings for large fires with 30 hotspots) are compelling, with visualizations (e.g., heatmaps in Figures 3-4) aiding interpretability. The custom simulator, grounded in established models (WRF-Fire, Hansen's extinguisher efficacy), adds credibility.
3. The approach is versatile for dynamic domains like urban firefighting, medical rescue, or flood control, where pure RL fails due to variance. It emphasizes result-oriented learning without over-relying on end-to-end RL, potentially inspiring hybrid systems.

**Weaknesses:**

1. The paper justifies avoiding SMAC or other MARL benchmarks due to their focus on small-scale competitive games, but this limits comparability. Custom forest fire scenarios, while realistic, may be tailored to favor the method (e.g., homogeneous drones suit shared policies). Comparisons to human firefighters feel mismatched—drones have advantages like aerial mobility but ignore real-world constraints (e.g., battery life, regulatory issues). Traditional MARL baselines are tested with only 25 agents, potentially understating their performance in scaled-down settings.
2. Propositions and proofs are concise but overly assumptive.

**Questions:**

1. The custom simulator and drones are detailed in the appendix, but code isn't mentioned (though reproducibility statement claims full details). Hyperparameters (e.g., PPO clip ϵ=0.1, α=2500/β=3500 in rewards) are specified, but sensitivity analyses are missing. Real-drone tests are limited to 3 units coexisting with simulations, raising questions about full-scale physical deployment (e.g., communication delays, sensor noise).

---

> ### Author Response · Authors · 2025-11-21
> **Author Response #1, Part 1/3: Questions for Reviewer ZeLm**
>
> **AUTHOR RESPONSE PART 1/3:  QUESTIONS**
>
> We thank the reviewer for their insightful feedback. We are encouraged that they agree that "1. The framework addresses a critical gap … inspiring hybrid systems." We will address their questions below and plan to update the paper upon receiving the reviewer's feedback.
>
> >1 a) The custom simulator and drones are detailed in the appendix, but code isn't mentioned (though reproducibility statement claims full details).
>
> The scope of this paper is a novel approach to enable groups of homogeneous agents to autonomously learn to perform unpredictable tasks, including those with a massive state space not feasible with today's state-of-the-art approaches. Using drones to fight wildfires is used as an exemplary domain to showcase the effectiveness of this approach. We believe the proposed approach disclosed in Section 3, alongside the details relevant to the implementation of this proposed approach with the simulator, virtual, and real drones in the appendix, clearly provides a precise and detailed view of reproducing the research and its results. We can further clarify any implementation details or describe any components requested by the reviewer.
>
> >1 b) Hyperparameters (e.g., PPO clip $ \epsilon $=0.1, $ \alpha $=2500/$ \beta $=3500 in rewards) are specified, but sensitivity analyses are missing.
>
> The main goal of experiments is to use an exemplary fire-fighting domain to test aspects of the two-phase approach. Sensitivity tests would instead address specific algorithms of the exemplary domain without addressing a specific aspect of the two-phase approach, and hence were not disclosed. Each algorithm in Figure 6 was evaluated with an optimal set of hyperparameters obtained after testing for these cases. In our pursuit for the best hyperparameters, we evaluated algorithms of our approach for different hyperparameters, and we can include such analysis if recommended by the reviewer, such as for the top-performing two-phase PPO. Specific to our PPO testing, a clip value of 0.1 provided sufficient policy updates while maintaining stability. The unpredictability of the environment causes a large PPO clip value, like 0.2-0.3, to lack stability, resulting in ineffective policy refinement. Variables $ \alpha $ and $ \beta $ represented as 2500 and 3500, providing a greater emphasis on area reduction over intensity reduction.
>
> >1 c) Real-drone tests are limited to 3 units coexisting with simulations, raising questions about full-scale physical deployment (e.g., communication delays, sensor noise).
>
> The scope of this paper is a novel approach to enable groups of homogeneous agents to autonomously learn to perform unpredictable tasks, including those with a massive state space not feasible with today's state-of-the-art approaches. We use the exemplary firefighting domain to demonstrate various aspects of our novel approach. The real drones, virtual drones, and associated controls constitute an effective testbed, but are not meant for a full-scale real-world deployment. Such deployment would involve, among other things, significant hardware improvements, including high-capability infrared, rangefinding, precise RTK location, and high-capacity water and extinguishing gear.
>
> The custom test environment was specifically designed to evaluate scalability issues with many agents. Same codebase between virtual and real drones enables them to coexist with real drones using camera and sensors against sampling of these inputs from fire datasets, along with simulated activities for virtual drones. Real drones operate alongside virtual or other real peers, all coexisting under a common custom ground control implementation that launches virtual drones with special virtual settings. This design allows us to test performance in the presence of a very large number of agents with real messages, unearthing any scalability issues and communication delays that would be encountered with a very large number of real drones, using the test setup with many virtual drones. The purpose of such a hybrid setup was also to visualize how real drones perform their activities in the presence of a very large number of other real/virtual drones. Sensor noise issue is not expected to be a major concern as phase-1 pipeline localizes it within a single real drone - it uses camera and sensor readings to derive the hotspot and edges, which are then propagated for holistic task decomposition and activity assignments. The shared policy-based phase-2 helps eliminate the bulk of MARL coordination issues, leading to massive scalability. We can include this discussion in the final version.

---

> ### Author Response · Authors · 2025-11-21
> **Author Response #1, Part 2/3: Addressing Weakness 1 for Reviewer ZeLm**
>
> **AUTHOR RESPONSE PART 2/3:  ADDRESSING WEAKNESS 1**
>
> We thank the reviewer for their insightful feedback. We are encouraged that they agree that "1. The framework addresses a critical gap … inspiring hybrid systems." We will address the weaknesses below and plan to update the paper upon receiving the reviewer's feedback.
>
> >1. a) The paper justifies avoiding SMAC or other MARL benchmarks due to their focus on small-scale competitive games, but this limits comparability. Custom forest fire scenarios, while realistic, may be tailored to favor the method (e.g., homogeneous drones suit shared policies).
>
> The main idea of the research is scalable MARL, and since SMAC did not offer robust testing beyond 30 agents, we could not test most aspects of our proposed approach. Absence of any other SOTA benchmark for this pioneering area required us to build a custom testbed to effectively test our paradigm. Our testbed not only tested our paradigm but also presented a comparative analysis with the SOTA algorithms.
>
> We respectfully disagree with the assessment of "tailored to fit the method." This paper focuses on a novel approach that expands MARL to massive collaborative tasks found in real-world environments, so it's necessary to evaluate the approach using a test method capable of evaluating various aspects of such a domain. As introduced in section 3.1, the system comprises a set of homogenous groups, and sharing policies across these agents in these groups is an inherent aspect of our paradigm. Having different groups of homogeneous agents in a massively scalable solution is not unusual for such real-world tasks, and in most cases a pragmatic reality of maintaining a large fleet of agents. As disclosed in section 4.1, our test methodology is not tailored to maximize evaluation returns or bias any results for this paradigm.
>
>
> We initially used SMACv2 for ideation of our two-phase approach and to compare it with QMIX and MAPPO. Scalability issues caused us to stop using SMACv2. Nevertheless, SMACv2 testing provided some key insights that we can share in the final paper. Of particular interest was the ability to use Phase-1 to significantly expedite learning by targeting it for a specific target scenario. Phase-1 enabled pruning the RL search-space, and we tested a strategy for ranged unit-types and a hybrid strategy for ranged, melee, and exploder unit-types. Phase-1 prunes the agent's search space using specific patterns based on field unit-types and their actions, and phase-2 quickly learns actions in this smaller space, beating the SOTA algorithms. However, the phase-1 strategy addressed only its specific scenario. E.g., the phase-1 ranged-only strategy worked for the ranged scenarios but not for the hybrid scenarios - a phase-1 hybrid strategy was required for the hybrid scenarios. These results showed two-phase approach trained much faster and performed slightly better than QMIX and MAPPO in the long run. By targeting specific aspects of the problem domain, it quickly mastered critical patterns and learned to ignore the vast non-essential details that explode the search space. We can share our results in an additional appendix and their impact on our proposed approach in the final version.
>
> >1 b) Comparisons to human firefighters feel mismatched—drones have advantages like aerial mobility but ignore real-world constraints (e.g., battery life, regulatory issues).
>
> We agree with the reviewer regarding the advantages and constraints of drones. Our goal is to use the exemplary system to strictly evaluate the effectiveness of our approach in using MARL, and therefore, we compare how drones' fire-fighting actions fare against well-established conventional fire-fighting. Our intention is not to demonstrate a full-fledged real-world fire-fighting system deployment. Such a system would certainly need to consider issues like battery life, regulatory issues, and many other practical concerns.
>
> >1 c) Traditional MARL baselines are tested with only 25 agents, potentially understating their performance in scaled-down settings.
>
> As shown in Figure 8, traditional MARL faltered beyond 30 agents, and as no reasonable testing was feasible beyond 25 agents, these were not understated in scaled-down settings.

---

> ### Author Response · Authors · 2025-11-21
> **Author Response #1, Part 3/3: Addressing Weakness 2 for Reviewer ZeLm**
>
> **AUTHOR RESPONSE PART 3/3:  ADDRESSING WEAKNESS 2**
>
> >2. Propositions and proofs are concise but overly assumptive.
>
> We believe that our assumptions reflect the practical considerations for the proposed muti-agent reinforcement learning paradigm. The analysis relies on three assumptions, each of which aligns with how large-scale multi-agent systems are actually deployed:
>
> 1) Homogeneous group for shared policy (Proposition 1).
> Our framework does not assume global homogeneity. Instead, policies are used within smaller groups of homogenous agents, grouped by similar sensing, actuation, and capability profiles. This is consistent with real-world deployments, where fleets naturally consist of classes of similar types and structures of drones. If a group exhibits internal heterogeneity, it can be further subdivided - our framework imposes no restriction on the number of groups - until this homogeneity is met. We can clarify this explicitly in Theorem 1.
>
> 2) Sufficient/comprehensive task decomposition (Proposition 2, Theorem 1).
> Our decomposition assumption formalizes this practical goal of full agent utilization, where a task is decomposed into activities such that these activities are assigned to the best capable agent, resulting in the highest possible utilization across the agents. We believe this is necessary in practice, as an activity lacking agents with the required capabilities will not be performed effectively, and agents not assigned sufficient activities lead to underutilization of the available capability.
>
> 3) Performance improvement under experience merging (Proposition 2).
> Merging additional trajectories improves the shared policy by incorporating information that helps it adapt. Only trajectory sets whose update direction increases expected performance are retained; those that would decrease performance are discarded. Thus, every update step moves the policy in an improving direction, ensuring the process is not assumptive but explicitly performance-aligned.
>
> Taken together, these assumptions are not overly strong but rather mirror the operational structure of real multi-agent systems: agents grouped by capability, tasks matched to those capabilities, and shared experience improving policy quality within a group. We can make additional updates in the proofs to make these clearer, as well as clarify any other assumptions as requested by the reviewer.

---

> > ### Comment · Reviewer_ZeLm · 2025-11-26
> >
> > I have read the author's response and other reviewers' opinions. Currently, I have no further questions.

---

> ### Author Response · Authors · 2025-12-03
> **Author Response #2, for Area Chair and Reviewer ZeLm**
>
> We thank the reviewer for their thoughtful feedback and are grateful that, after reviewing our clarifications and experimental additions, they had revised their score from 6 to 8. In response to their concerns about SMAC and assumptions, we addressed those issues by providing detailed results and clarification for the assumptions. We also addressed sensitivity and test scalability in our official comments, which gave a positive impression resulting in an improvement in their rating.
> This increase directly reflects that the questions raised were fully addressed and that the improvements strengthened the contribution and clarity of the work. We respectfully ask the area chair to take the strong technical merits and completeness of our research that contributed to this positive rating adjustment into consideration. It also demonstrates that engagement with reviewer feedback substantially improved confidence in the paper’s quality and significance for the community.
>
> Based on the reviewer's comments, we have added clarifications and made refinements to the final version of the paper as described below:
>
> **Questions**:
> >1a. The custom simulator and drones are detailed in the appendix, but code isn't mentioned (though reproducibility statement claims full details).
>
> We have provided comprehensive details regarding simulation-based and drone-based testing throughout the paper. As the reviewer agreed, our clarification with the reviewer was met, and the reviewer's updated rating signifies adequate disclosure and reproducibility.
>
> >1b. Hyperparameters (e.g., PPO clip ϵ=0.1, α=2500/β=3500 in rewards) are specified, but sensitivity analyses are missing.
>
> As clarified in update to Appendix A.3.2: [Lines 972-983] "The primary purpose of experimentation was to evaluate the two-phase approach using an exemplary fire-fighting domain… resulting in greater refinement efficacy."
>
> >1c. Real-drone tests are limited to 3 units coexisting with simulations, raising questions about full-scale physical deployment (e.g., communication delays, sensor noise).
>
> - As clarified in update to Appendix A.3.2: [Line 923-932]: "The custom test environment was specifically designed to evaluate scalability issues with many agents…very large number of other real/virtual drones.
> - Moreover, as also clarified in Appendix A.3.2: [Line 936-940]: "Note that the scope of this paper is a novel approach to enable groups of homogeneous agents…associated controls constitute an effective testbed."
>
> **Weaknesses**:
> >1a. The paper justifies avoiding SMAC or other MARL benchmarks due to their focus on small-scale competitive games, but this limits comparability. Custom forest fire scenarios, while realistic, may be tailored to favor the method (e.g., homogeneous drones suit shared policies).
>
> We clarified this in update to Appendix A.5: [Lines 1050-1187]. We have included a very detailed disclosure of a completely new section for the SMAC v2-based comparison of the two-phase strategy with QMIX and MAPPO that was used in the early stages of this research to help with the ideation behind this research, and detailed disclosure of our insights and issues.
>
> >1b. Comparisons to human firefighters feel mismatched—drones have advantages like aerial mobility but ignore real-world constraints (e.g., battery life, regulatory issues).
>
> As clarified in update to Appendix A.3.2: [Line 924-940] "Same codebase between virtual and real drones enables them to coexist with real drones… associated controls constitute an effective testbed for testing these aspects."
>
> >1c.Traditional MARL baselines are tested with only 25 agents, potentially understating their performance in scaled-down settings.
>
> As clarified in update to Appendix A.3.6: [Line 999-1000]: "It must be noted that since traditional MARL does not scale well beyond around 30 agents, the tests were conducted with only two-phase algorithms beyond 30 agents."
>
> >2.Propositions and proofs are concise but overly assumptive
>
> As clarified in update to Appendix A.1.5: [Line 715-735]: We have added the following clarification in Appendix 1.5:  "We believe that our assumptions reflect the practical considerations for the proposed multi-agent reinforcement learning paradigm. The analysis relies on three assumptions…mirror the operational structure of real multi-agent systems."
>
> In response to the reviewer's comment, we added a new section to our paper "A.1.5 Practical Considerations", which discusses how our theoretical results match practical considerations posed by real-world MARL systems. This includes our assumptions of homogeneity within each policy group, sufficient/comprehensive task decomposition, and performance improvement under experience merging.
>
> We thank this reviewer for their insightful comments and the area chair for their consideration.

---

### Official Review · Reviewer_gFNh · 2025-10-31

**Soundness:** 2
**Presentation:** 3
**Contribution:** 2
**Rating:** 4
**Confidence:** 4

**Summary:**

To address large-scale, dynamic, and unpredictable real-world tasks, this paper proposes a two-stage iterative multi-agent reinforcement learning framework that alternates between a Refocus and Refine process. In each iteration, the Refocus phase determines the current optimal task allocation scheme, while the Refine phase leverages collective intelligence for optimizing execution and policy learning at the agent level. The proposed framework is validated through realistic firefighting simulations with drone swarms, demonstrating strong empirical performance.

**Strengths:**

1. The paper features real-world experiments with substantial engineering achievements — a type of contribution that is relatively rare in the ICLR community.

2. It represents a valuable attempt to apply MARL methods in real-world scenarios, bridging the gap between theoretical frameworks and practical deployment.

**Weaknesses:**

1. Outdated Literature Review

The literature discussed in the Related Work section is largely outdated, with many works from four or five years ago. I understand that the authors may have chosen representative or characteristic studies to highlight the novelty of their own work, but the selected references do not reflect the current state of research, especially in Hierarchical Reinforcement Learning and Task Partitioning & Role Assignment.
Recent works relevant to dynamic and non-stationary task environments include (but are not limited to):

Hierarchical Reinforcement Learning (HRL):

[1] End-to-end hierarchical reinforcement learning with integrated subgoal discovery. IEEE TNNLS, 2021.

[2] Hierarchical reinforcement learning for non-stationary environments. SSCI, 2023.

[3] Exploring the limits of hierarchical world models in reinforcement learning. Scientific Reports, 2024.

[4] Hierarchical reinforcement learning with uncertainty-guided diffusional subgoals. ICML, 2025.

Task Partitioning & Role Assignment:

[5] Ldsa: Learning dynamic subtask assignment in cooperative multi-agent reinforcement learning. NeurIPS, 2022.

[6] Learning to transfer role assignment across team sizes. AAMAS, 2022.

[7] Dynamic role discovery and assignment in multi-agent task decomposition. Complex & Intelligent Systems, 2023.

[8] Automatic grouping for efficient cooperative multi-agent reinforcement learning. NeurIPS, 2023.

[9] Dynamic subtask representation and assignment in cooperative multi-agent tasks. Neurocomputing, 2025.

2. Clarity of Notation and Contextualization

Section 3.1 provides a clear and rigorous formal setup with well-defined notation. However, I strongly recommend that the authors illustrate the meaning of each symbol using the forest firefighting example, to help readers understand the motivation and reasonableness behind each assumption. The current version, while clearly written, still leaves several conceptual ambiguities (see “Questions” section below).

3. Figure 1 is Confusing

As the core figure of the paper, Figure 1 is confusing and fails to clearly depict the two phases of the framework. The caption actually conveys more clarity than the figure itself. A redesign that visually differentiates the Refocus and Refine processes is highly recommended.

4. Limited Methodological Novelty

The paper reads more like a technical report than a methodological contribution. Essentially, it presents an intuitive framework rather than a novel algorithmic method. While proposing a framework is valuable, most components in the experiments are based on or quite similar to existing methods, and the analysis largely explores their behavior within the proposed setup. Overall, this work is stronger in engineering and application than in methodological innovation.

**Questions:**

1. In $A_g$, there is no explicit dependence on $t$. Does this mean $A_g$ does not vary over time?

2. Can two groups $A_{g_i}$ and $A_{g_j}$ simultaneously handle overlapping subsets of activities?

3. Could the simultaneous partitioning of both tasks and agents potentially reduce overall efficiency? For instance, might some agents need to wait for others to finish certain sub-tasks before proceeding?

4. The constraint that “each group’s number of activities should be no less than the number of agents in that group” seems overly strong. With a large number of agents, can this always be satisfied? Does it imply that each agent performs multiple sub-tasks at every time step $t$? If so, how is this feasible, given that an agent’s action $u_i$ might benefit some sub-tasks but hinder others?

5. Are $r(w^t_j)$ and $\kappa(a_i)$ prior knowledge or expert-defined heuristics?

6. How are real drones and simulated drones coexisting in the same environment (line 300)? The description seems inconsistent: at one point (line 320) you mention 3,000 simulated drones, while later (line 339) you refer to 25 agents in a MARL comparison. Could you clarify this setup?

7. Is drone capability evaluated solely based on fire-extinguishing capacity?

8. You mentioned testing the framework on several MARL benchmarks. Could you provide detailed results on standard benchmarks such as SMAC? This is crucial to assess the framework’s potential impact on future MARL research. I genuinely appreciate the engineering effort and real-world implementation, but practical success alone does not establish superiority. Demonstrating that existing MARL algorithms perform better under your proposed framework on well-known benchmarks like SMAC or GRF would greatly strengthen your contribution. Without such evidence, I am currently leaning toward a negative evaluation. However, if the authors can show competitive or superior benchmark performance, I would be delighted to see this work presented to the ICLR community.

---

> ### Author Response · Authors · 2025-11-21
> **Author Response #1, Part 1/2: Questions for Reviewer gFNh**
>
> **AUTHOR RESPONSE PART 1/2: QUESTIONS**
>
> We thank the reviewer for their insightful feedback. We are encouraged that they agree that "The paper features… practical deployment." We address their questions and plan to update the paper upon receiving the reviewer's feedback.
>
> >1. In $A_g$, … time?
>
> As disclosed in Section 3.1, the grouping of agents $\mathcal{A_g}$ is done based on their inherent capabilities to perform the task $\mathcal{W}(t) $: agents in $\mathcal{A}_{g}$ are homogeneous and that does not change over time. The number of agents in a specific group may change as agents join or leave, e.g., to refuel, but agents don't switch groups. We can clarify this in the final version.
>
> >2. Can two … activities?
>
> We appreciate the reviewer for this insight. Yes, assignment matrix $X_t$ represents agent $a_i$ 's assignment to activity $w_{jt}$. Agent's group does not constrain its activity assignment. $r(w_{jt})$ is the minimum agents to execute $w_{jt}$, and when $r(w_{jt})$ = 2, two or more agents, possibly from different agent groups, may be assigned to $w_{jt}$. Thus two groups may have agents working on overlapping subset of activities.  We can clarify this in the final version.
>
> >3. Could the … proceeding?
>
> No, an agent's task assignment may be continuously revised and is not tied to activity completion - we iterate continuously between the two phases. Phase-1 holistic view of the current task $\mathcal{W}(t) $ governs such partitioning, and optimizing activity to agent assignment leads to tremendous gains.  We can clarify this in the final version.
>
> >4. The constraint … others?
>
> The idea is for decomposition to target full agent utilization so that all available agents remain active. So  $\mathcal{D}$ partitions the task, assigning activities to agents based on their capability. With excess activities, some activities won't get assigned immediately. If there are excess agents, some agents remain idle - try to avoid this by attempting $|\mathcal{W_g}(t)| \geq |\mathcal{A}_{g}|$: this is attempted, but it is not a requirement. We can clarify this in the final version.
>
> >5. Are $r({w_j}^t)$ … heuristics?
>
> Depending on the task, these metrics are typically prior domain knowledge specified by the expert or predefined while creating the domain.  We can clarify this in the final version.
>
> >6. How are … setup?
>
> Same codebase between virtual and real drones enables them to coexist with real drones using camera and sensors against sampling of these inputs from fire datasets, along with simulated activities for virtual drones. Real drones operate alongside virtual or other real peers, all coexisting under a common custom ground control implementation that launches virtual drones with special virtual settings.
>
> The testing environment can simulate more than 3000 drones, and two-phase algorithm testing shows effective scaling beyond 1000 agents. To compare two-phase algorithms against SOTA MARL algorithms, we had to limit test comparisons to only 25 agents, as SOTA MARL algorithms fail beyond 25 agents as seen in Figure 8. We can clarify this in the final version.
>
> >7. Is drone …  capacity?
>
> Drone capability is based on domain-specific activities. The exemplary fire-fighting domain includes aspects of this domain, including fire-extinguishing capacity, fire-extinguishing type, and drone speed.
>
> >8. You …  ICLR community.
>
> We thank the reviewer for this insight. We initially used SMACv2 for ideation of our two-phase approach and compared it with QMIX and MAPPO. However, scalability issues beyond ~25 agents made it unsuitable for our needs. Other benchmarks had similar limits, so we adopted an exemplary firefighting evaluation that enabled deeper analysis and a real-world demonstration.
>
>
> Nevertheless, SMACv2 testing provided some key insights that we can share in the final paper. Of particular interest was the ability to use Phase-1 to significantly expedite learning by targeting it for a specific target scenario. Phase-1 enabled pruning the RL search-space, and we tested a strategy for ranged unit-types and a hybrid strategy for ranged, melee, and exploder unit-types. Phase-1 prunes the agent's search space using specific patterns based on field unit-types and their actions, and phase-2 quickly learns actions in this smaller space, beating the SOTA algorithms. However, the phase-1 strategy addressed only its specific scenario. E.g., the phase-1 ranged-only strategy worked for the ranged scenarios but not for the hybrid scenarios - a phase-1 hybrid strategy was required for the hybrid scenarios. These results showed two-phase approach trained much faster and performed slightly better than QMIX and MAPPO in the long run. By targeting specific aspects of the problem domain, it quickly mastered critical patterns and learned to ignore the vast non-essential details that explode the search space. We can share our results in an additional appendix and their impact on our proposed approach in the final version.

---

> ### Author Response · Authors · 2025-11-21
> **Author Response #1, Part 2/2: Addressing Weaknesses for Reviewer gFNh**
>
> **AUTHOR RESPONSE PART 2/2:  ADDRESSING WEAKNESSES**
>
> We thank the reviewer for their insightful feedback. We are encouraged that they agree that "The paper features real-world … practical deployment." We will address the weaknesses below and plan to update the paper upon receiving the reviewer's feedback.
>
> >1. Outdated Literature … Neurocomputing, 2025.
>
> We appreciate the reviewer for this helpful information. This would be a great addition to the final version of our paper.
>
> >2. Clarity of Notation …  section below).
>
> We sincerely apologize for the conceptual ambiguities regarding this section of the paper. We can translate the formal setup for the exemplary fire-fighting domain in the appendix of the final version.
>
> >3. Figure 1 is … recommended.
>
> We can revise our Figure 1 to visually illustrate the purpose of each phase in a different manner that provides more details of what happens in each phase.
>
> >4. Limited Methodological … innovation.
>
> We appreciate the reviewer for appreciating our strong engineering and application, which shows a unique way of using a massively scalable MARL paradigm to solve some tough real-world problems. However, in Section 3, we disclose a detailed method that enables us to scale significantly more than the 30 agents possible with SOTA algorithms using a concrete methodological innovation that not only makes this possible but does so by enabling the use of many existing proven RL and MARL algorithms - something that is novel and makes this approach highly versatile. Aspects of this methodology are also outlined in Algorithms 1 and 2 in Section 3.2 and Algorithms 3 and 4 in Appendix A.2.
>
> a) Phase-1: Using a holistic task analysis, decompose and optimally assign activities to groups of homogeneous agents. Pruning the RL search space in this manner is novel and instrumental to enabling MARL for such complex domains.
>
> b) Phase-2: Using a shared policy, learn to perform an activity effectively, and record a trajectory for this activity. This greatly reduces the coordination overheads in a novel manner, which further enables MARL scalability, while also enabling leveraging robust existing RL/MARL algorithms.
>
> c) Share experience learning using these trajectories. The number, selection, and merging of these trajectories is also a unique aspect of our methodological innovation.
>
> Iterating these steps leads to continuous and automatic evolution across a very large number of agents. Therefore, we believe this work is based on a sound new method that pioneers in achieving MARL scalability and solving complex real-world problems that were not possible before. Therefore, we believe we have disclosed a pragmatic and versatile methodological innovation.

---

> ### Author Response · Authors · 2025-12-03
> **Author Response #2 for Area Chair and Reviewer gFNh**
>
> We thank the reviewer for their constructive comments and for expressing that they would be **delighted to see our work presented to ICLR community** if strong SMAC v2 benchmarking were provided. In response, we have added full SMAC v2 experimental results demonstrating competitive and superior performance relative to MAPPO and QMIX that were already performed earlier. This evaluation directly reflects the reviewer's appreciation for our work and the impact of enabling this novel, scalable MARL solution to solve real-world problems that are not possible today. We respectfully request that the area chair take this into account in the final decision.
>
> Based on the reviewer's comments, we have added clarifications and made refinements to the final version of the paper as described below:
>
> **Questions**:
>
> Questions 1-7: We added a new section in the paper "A.3.2 Agents, Activity Assignments, and Execution for Exemplary System". This section addresses these questions, as well as elaborates further on how it applies to the exemplary domain.
>
> >1. In $ A_g $, there is no explicit dependence on $ t $. Does this mean $A_g$ does not vary over time?
>
> As clarified in update to Appendix A.3.2: [Line 892-894] "A drone may temporarily … is based on its capabilities."
>
> >2. Can two groups $A_{g_i}$ … subsets of activities?
>
> As clarified in update to Appendix A.3.2: [Line 901-905] "An activity may require one… have agents working on an overlapping subset of activities."
>
> >3. Could the … sub-tasks before proceeding?
>
> As clarified in update to Appendix A.3.2: [Line 910-917] "An agent is not required to operate on its assigned activity to completion… be assigned to any agent."
>
> >4. The constraint … but hinder others?
>
> As clarified in update to Appendix A.3.2: [Line 910-917] "If the task is decomposed such that … $|\mathcal{W_g}(t)| \geq |\mathcal{A_g}|$."
>
> >5. Are $ r({w_j}^t) $ and $ k(a_i) $ prior knowledge or expert-defined heuristics?
>
> As clarified in update to Appendix A.3.2: [Line 905-909]"The $r(w_{jt})$ and $\kappa(a_i)$ are typically prior domain knowledge specified by the expert or pre-defined while creating … handle many small fire edges."
>
> >6. How are … this setup?
>
> As clarified in update to Appendix A.3.2: [Line 923-935] "The custom test environment … failed beyond 25 agents."
>
> >7. Is drone capability evaluated solely based on fire-extinguishing capacity?
>
> As clarified in update to Appendix A.3.2: [Line 888-892] Agents are categorized into groups based on their capabilities, … 100 liters respectively.
>
> >8. You mentioned … However, if the authors can show competitive or superior benchmark performance, I would be delighted to see this work presented to the ICLR community.
>
> As clarified in update to Appendix A.5: [Lines 1050-1187]: We have included a very detailed disclosure of a completely new section for the SMAC v2-based comparison of the two-phase strategy with QMIX and MAPPO that was used in the early stages of this research to help with the ideation behind this research, and a detailed disclosure of our insights and issues.
>
> We would like to bring **strong attention** to the reviewer's statement that "However, if the authors can show competitive or superior benchmark performance, I would be delighted to see this work presented to the ICLR community."
>
> We **not only disclosed such work that clearly shows our superior benchmark performance** in Section A.5. We also disclosed a detailed discussion of how such work led to the ideation behind our research and how we progressed beyond limitations of current test frameworks and SOTA algorithms. Unfortunately, due to our inability to engage with the reviewer on the subject, we believe we are being denied an irrefutable opportunity to change this reviewer's rating of our research. We urge the area chair to earnestly consider this fact in granting us consideration for a **highly favorable rating** for this reviewer
>
> **Weaknesses**:
> >1. Related Works:
>
> As clarified in Section 2 and bibliography, we have included the reviewers recommended related works.
>
> >2: Clarity of Notation and Contextualization Section 3.1… still leaves several conceptual ambiguities (see “Questions” section below).
>
> As clarified in Section A.3.2: We have added this whole new section that provides an informal discussion with the exemplary system along with relevant examples to help deepen the reader's understanding of these concepts.
>
> >3. Figure 1 is …  is highly recommended.
>
> We have replaced Figure 1 with a more detailed figure that accurately portrays the critical aspects of the two-phase paradigm.
>
> >4. Limited Methodological … in methodological innovation.
>
> We have provided a detailed response on the subject in our original comment that clearly highlights the methodological innovation of our research.
>
> We thank this reviewer for their insightful comments and the area chair for their consideration.

---

### Official Review · Reviewer_2oXC · 2025-11-01

**Soundness:** 3
**Presentation:** 3
**Contribution:** 3
**Rating:** 6
**Confidence:** 2

**Summary:**

This paper introduces a two-phase framework to improve the scalability of multi-agent reinforcement learning in large, dynamic environments. The key idea is to decouple global coordination and local learning: the system first decomposes and redistributes tasks among homogeneous agent groups, then each group learns and refines activity-specific policies through shared experience. By iteratively alternating between these phases, the framework enables adaptive, efficient cooperation across many agents.

**Strengths:**

- The paper presents a clear 2-stage framework that separates global task allocation from local policy learning, effectively improving scalability in large multi-agent systems.

- The iterative design enables continuous adaptation to dynamic environments, the performance seems to be better than the baseline.

**Weaknesses:**

- The task decomposition process relies heavily on heuristic and domain-specific components, limiting generality and reproducibility.

- Theoretical guarantees are based on strong assumptions (e.g., agent homogeneity and sufficient decomposition) that are generally not met in practice.

**Questions:**

1. The theoretical results rely heavily on the assumption that agents are perfectly homogeneous, making their trajectories fully interchangeable. In realistic systems, agents often differ in sensing, actuation, or local context—how would the proposed merging and shared policy updates behave when this assumption is relaxed?

2. The framework assumes that shared experience merging among agents improves learning efficiency, but for inhomogeneous agents this could introduce bias or destabilize training. Has the method been tested or theoretically analyzed under partially heterogeneous conditions, and what mechanisms (e.g., importance weighting or representation alignment) might mitigate these effects?

---

> ### Author Response · Authors · 2025-11-21
> **Author Response #1, Part 1/2: Questions for Reviewer 2oXC**
>
> **AUTHOR RESPONSE PART 1/2: QUESTIONS**
>
> We thank the reviewer for their insightful feedback. We are encouraged that they agree that "The paper presents a clear 2-stage … seems to be better than the baseline." We will address their questions below and plan to update the paper upon receiving the reviewer's feedback.
>
> >1. The theoretical results rely …  this assumption is relaxed?
>
> We thank the reviewer for this comment. We agree that a system with all homogeneous agents is not practical and therefore, as introduced in section 3.1, our approach uses **a set of homogeneous groups.** Sharing policies across agents within each group is an inherent aspect of our paradigm. Large-scale MARL collaboration between groups of homogeneous agents is a major innovation of this paper and a pragmatic reality of realizing such scalable solutions.
>
> Differences in sensing for phase-1 are less of a concern as phase-1 pipeline infers and reports edges and hotspots, and such inference is quite tolerant to sensing deviations. Sensing and actuation for phase-2 operations in executing policy-driven actions and trajectories could also experience such deviations. However, these  phase-2 operations do not suffer from variations in sensing or actuation as much as the operations are specifically categorized to operate within a specific range, as determined by the capabilities defining the homogeneous agent set. In rare cases, when very sensitive operations are not tolerant to a single range, it would split a homogeneous group into multiple groups depending on the tolerated multiple ranges for such operations. This allows phase-2 sharing of policies across agents with similar capabilities, overcoming variations in sensing, actuation, or local context. We can include this discussion in the final version.
>
> >2. The framework assumes that … mitigate these effects?
>
> We thank the reviewer for this comment. Our approach fully recognizes that not all agents in the system can be homogeneous. As introduced in section 3.1, a set of homogeneous groups is used, and sharing policies across homogeneous agents within such a group allows policy-based operations and trajectory learning. For most massively scalable solutions, sizable groups of homogeneous agents are a pragmatic reality of maintaining a large fleet of agents. There is no inherent limitation to the number and size of such agent groups and therefore, undesired bias or destabilized training are not encountered across the homogeneous agents within such groups. As agents do not operate outside such groups, we do not encounter this issue. Moreover, shared experience learning is tolerant to deviations customary across agents within the homogeneous groups. We have evaluated the method under partially heterogeneous conditions, where different groups vary in size (small/medium/large), role (hotspot vs. edge suppression), and operational context. Because each group maintains its own specialized policy and only merges experiences internally, we do not observe any destabilization. We can include this discussion in the final version.

---

> ### Author Response · Authors · 2025-11-21
> **Author Response #1, Part 2/2: Addressing Weaknesses for Reviewer 2oXC**
>
> **AUTHOR RESPONSE PART 2/2: ADDRESSING WEAKNESSES**
>
> We thank the reviewer for their insightful feedback. We are encouraged that they agree that "The paper presents a clear 2-stage  … seems to be better than the baseline." We will address the weaknesses below and plan to update the paper upon receiving the reviewer's feedback.
>
> >1. The task decomposition … and reproducibility.
>
> We thank the reviewer for recognizing that phase-1 of our approach can take advantage of heuristic and domain-specific components, as demonstrated in the exemplary system disclosed in the paper. Although such optimizations greatly boost the efficacy of our approach, phase-1 is about holistic refocusing of the task, where a generic task decomposition and agent assignment still allows this paradigm to operate without limiting generality or reproducibility.
>
> We believe this to be a big strength of our approach, as it also provides a means to inject prevalent external/domain knowledge in the learning process, making it feasible to significantly prune the massive search space. Figure 8 shows how the current state-of-the-art MARL approaches can't scale for such a massive state space, limiting their real-world MARL applicability. Most MARL approaches fail to inject a means to curb exploring irrelevant portions of the massive search-space, resulting in their failure for pragmatic real-world use of such complex, huge tasks.
>
> All remaining components, including A* task assignment and the entirety of phase-2 "refine", are completely domain-independent. It is also possible to transfer optimizations similar to Phase-1 optimizations using sensory and image data, demonstrated in the exemplary system, to other domains. E.g., Locating fire-areas using transformer pipelines can be transferred to locating flooded areas for a system of autonomous robots in a flood-control application. Fire-fighting activities of the exemplary system are replaced by flood-control activities that robots learn in an identical manner for the flood-control application.
>
> Therefore, our iterative two-phase approach, possibly optimized by injecting additional knowledge/human-in-the-loop, does not limit generality. It is also possible to reproduce the behaviors, even when additional knowledge is injected, as demonstrated by results disclosed in Section 4.2 across multiple trials. We can highlight this critical aspect in the final version of our paper.
>
> >2. Theoretical guarantees … not met in practice.
>
> We believe that our assumptions reflect the practical considerations representative of the proposed multi-agent reinforcement learning system that are met in practice. The analysis relies on three assumptions, each of which aligns with how large-scale multi-agent systems are actually deployed:
>
> 1) Homogeneous group for shared policy (Proposition 1).
> Our framework does not assume global homogeneity. Instead, policies are used within smaller groups of homogenous agents, grouped by similar sensing, actuation, and capability profiles. This is consistent with real-world deployments, where fleets naturally consist of classes of similar types and structures of drones. If a group exhibits internal heterogeneity, it can be further subdivided - our framework imposes no restriction on the number of groups - until this homogeneity is met. We can clarify this explicitly in Theorem 1.
>
> 2) Sufficient/comprehensive task decomposition(Proposition 2, Theorem 1).
> Our decomposition assumption formalizes this practical goal of full agent utilization, where a task is decomposed into activities such that these activities are assigned to the best capable agent, resulting in the highest possible utilization across the agents. We believe this is necessary in practice, as an activity lacking agents with the required capabilities will not be performed effectively, and agents not assigned sufficient activities lead to underutilization of the available capabilities.
>
> 3) Performance improvement under experience merging (Proposition 2).
> Merging additional trajectories improves the shared policy by incorporating information that helps it adapt. Only trajectory sets whose update direction increases expected performance are retained; those that would decrease performance are discarded. Thus, every update step moves the policy in an improving direction, ensuring the process is not assumptive but explicitly performance-aligned.
>
> Taken together, these assumptions are not overly strong but rather mirror the operational structure of real multi-agent systems that is generally met in practice: agents grouped by capability, tasks matched to those capabilities, and shared experience improving policy quality within a group. We can make additional updates in the proofs to make these clearer, as well as clarify any other assumptions as requested by the reviewer.

---

> ### Author Response · Authors · 2025-12-03
> **Author Response #2 for Area Chair and Reviewer 2oXC**
>
> We thank the reviewer for their detailed questions and critiques, which helped refine and strengthen the paper.  In response to their suggestions, we addressed their questions pertaining to assumptions on agent homogeneity and trajectory merging, as well as concerns about domain-specificity. We believe these answers substantially improve the clarity and rigor of our work. We strongly believe that if granted the opportunity, the ability to communicate with this reviewer would have improved our rating for acceptance of our work. Therefore, we respectfully urge the area chair to consider this complete resolution of concerns as strong support for a favorable final decision.
>
> Based on the reviewer's comments, we have added clarifications and made refinements to the final version of the paper as described below:
>
> **Questions**:
> >1. The theoretical results rely heavily on the assumption that agents are perfectly homogeneous… behave when this assumption is relaxed?
>
> - As clarified in update to Section 3.2.2 [Line 178]: "Agents with same capabilities belong to a group \(A_g\), and these homogeneous agents share their experiences to collectively refine $\pi_{w_{jt}}$."
>
> - As also clarified in update to Appendix A.1.5: [Lines 718-722]: "Policies are used within smaller groups of homogenous agents, grouped by similar sensing, actuation, and capability profile…our framework imposes no restriction on the number of groups - until this homogeneity is met."
>
> >2. The framework assumes that shared experience merging among agents… mitigate these effects?
>
> As clarified in update to Appendix A.3.7: [Lines 1012-1015]: "Because each group of agents maintains its own specialized policy and only merges experiences internally, we do not observe any policy destabilization."
>
> **Weaknesses**:
> >1. The task decomposition process relies heavily on heuristic and domain-specific components, limiting generality and reproducibility.
>
> As clarified in update to Appendix A.4: [Lines 1031-1048]: "A unique aspect of this approach is the ability to complement pure reinforcement learning with adjunct strategies, including domain intelligence, learning algorithms, human-in-the-loop (HIL), to expedite learning…tackle complex tasks in dynamic and unpredictable environments."
>
> In response to the reviewer's comment, we added a new section to the paper "A.4 Injecting external domain/knowledge," which provides greater context into the scope and advantages of injecting external domain knowledge in addressing scalability and versatility.
>
> >2. Theoretical guarantees are based on strong assumptions (e.g., agent homogeneity and sufficient decomposition) that are generally not met in practice.
>
> As clarified in update to Appendix A.1.5: [Lines 715-735] "We believe that our assumptions reflect the practical considerations for the proposed multi-agent reinforcement learning paradigm. The analysis relies on three assumptions…mirror the operational structure of real multi-agent systems."
>
> In response to the reviewer's comment, we added a new section to our paper "A.1.5 Practical Considerations", which discusses how our theoretical results match practical considerations posed by real-world MARL systems. This includes our assumptions of homogeneity within each policy group, sufficient/comprehensive task decomposition, and performance improvement under experience merging.
>
> We believe these significant additions to our paper would serve to address the reviewer's concerns and would have likely led to an improvement in their rating. We thank this reviewer for their insightful comments and the area chair for their consideration.

---

### Official Review · Reviewer_19Yf · 2025-11-01

**Soundness:** 3
**Presentation:** 3
**Contribution:** 3
**Rating:** 8
**Confidence:** 3

**Summary:**

This paper introduces a two phase approach to solving multi-agent learning in complex unpredictable and dynamically changing environments.
The method assumes sets of homogeneous agents, with each set having a library of pre-trained policies.
Phase one involves using agents to gather information about the environment to use in a task distributor. The task distributor then decomposes the environment into sub-tasks, with the at least as many sub-tasks as there are agents to avoid wasted agent assignment. The tasks are then assigned to the agents.
Phase two involves each agent having been assigned a sub-task and chooses a policy from their library that is best for solving that task. The agents then perform their tasks and merge the best and worst experiences. Depending on the variation of the setups described, this might only use the best experiences or a weighted distribution over the experience. These are then used to refine the policies and store the updated in the policy bank. The policies are trained using PPO, DQN and/or AC methods.

They demonstrate this in a fire fighting scenario in which they have different classes of drones, small, medium, large. The simulations are derived from previous works which simulate the fire scenarios. This was simulated, and done with actual drones, with fire depicted by coloured fabric. (I have concerns for experiment with the actual drones that this might not follow a repeatable rule in updating the environment. Whereas the simulated ones are backed by data. The focus of the research appears to be on the algorithm and not the simulation however, so this is less of a concern for the purpose of the algorithm.)
They compare their method this to a base line of actual firefighters over various task sizes, varying the number of fire units and hotspots. They show that the algorithm outperforms the baseline by about 15-40%, which larger improvements on large task sizes.
They do an ablation on the task distribution with various setups, showing using a transformer and EdgeProgression performs the best.
They use three two-phase models: PPO, AC, DQN. All three beat traditional MARL methods: MAPPO, A2C and DQN. The two-phase PPO is shown to perform the best.
The weighted shared experience learning is shown to have the best performance.
They also show that the scalability of the method outperforms traditional MARL methods by a large margin.

Problem addresses: A scalable method to solving dynamic complex environments with multi-agent systems.

The proposed method/idea: A two phase approach for task assignment for different sets of homogeneous agents, and updated learning during the task.

The main claim is that they contribute to perform complex tasks with a large number of agents, and this is backed by empirical results as well as some theoretical results.

This work tackles an important problem in MARL scalability for complex and dynamic tasks.

Overall I believe this is good work. There are a few details about the experiment setup, particularly the real drones one, that could be expanded upon to make it clearer.

**Strengths:**

1) The paper clearly describes the methods.
2) The paper tests over a number of different methods and ablations.
3) The simulations are backed up by previous works.
4) The results suggest this method outperforms the baseline as well as different traditional MARL methods.
5) It appears to be a novel combination of task assignment and merged learning techniques to solve an important problem.

**Weaknesses:**

1) I may have missed something, but it is not entirely clear how the reward works, or how that is measured in the real life simulations. Perhaps expanding on this might help.
2) I'm not 100% certain about the baseline of the firefighters. I would love an elaboration on that.

3) Minor typos in the paper such as line 109 "the maximum numbers of activities than agent..."
I didn't track all of them, but a good read through should be done, as I did pick up a few others.

**Questions:**

1) I am assuming that agents cannot be reassigned tasks until it's current task has been completed? At least that's the understanding I got from the way you describe the tasks having time lengths. I ask because it seems like there could be a strange edge case where the task assignment phase could update a task to an agent and then reassign it again getting stuck in a loop if that is not the case.
2) With the policy library being updated, does this not introduce a non-stationarity issue, it seems the results don't suggest this to be a problem. It's a question out of curiosity.

---

> ### Author Response · Authors · 2025-11-21
> **Author Response #1 for Reviewer 19Yf**
>
> We thank the reviewer for their insightful feedback. We are encouraged that they agree that "The paper clearly describes  … solve an important problem." We will address the weaknesses below and plan to update the paper upon receiving the reviewer's feedback.
>
> **Questions:**
>
> 1. I am assuming … not the case.
>
> The two-phase approach iterates task assignment and execution over a specific time interval, configured depending on the task scale and type of domain. Reassignment is not mandated for each agent in every iteration, but a potential reassignment is possible, such as if an agent is better suited for a more optimal or new task activity, even if the current task hasn't been completed. If an ongoing task activity remains optimal for the agent, no reassignment is needed. Once a task activity is completed, it can no longer be assigned to any agent. Moreover, once a task activity is marked as assigned, there is no need to reassign it to an agent during the iteration, preventing any possibility of getting stuck in a loop.
>
> 2. With the policy … out of curiosity.
>
> We agree with the reviewer that real-time shared policy updates can introduce non-stationarity, since changing one agent’s policy alters the behavior that other agents observe. In many MARL systems, this creates a moving-target problem, as when all agents continually adapt to each other’s shifting policies, learning can destabilize.
>
> However, our approach doesn't cause immediate use of experience sharing, eliminating such potential non-stationarity.  Each agent executes real-time policy updates on its own through the operateAgent algorithms, refining the policies in real-time during firefighting tasks. Upon designated intervals, completion of a task, or reassignments, optimal trajectories from homogenous agents are merged to refine the policy bank. By deferring policy-bank updates and maintaining static policies during an execution window, the system avoids the primary mechanism that causes non-stationarity.
>
> **Weaknesses:**
>
> 1. I may have … this might help.
>
> As shown in Section 4.1 Lines 304-309, the reward function is shown as $R_a = \alpha \cdot min(\frac{\Delta I}{I}, k_1) + \beta \cdot min(\frac{\Delta A}{A}, k_2)$. This function is proportional to the change in intensity ($\frac{\Delta I}{I}$) and fire area ($\frac{\Delta A}{A}$), while the clipping terms $k_1$​ and $k_2$​ cap extreme values to prevent instability. It shows that a greater reward results from a greater reduction in fire intensity and area. Furthermore, the constants $\alpha$ and $\beta$ reflect the weightage of the intensity and area reduction in the reward, respectively, with sample values of 2500 and 3500 providing a greater emphasis on area reduction in order to prioritize the usage of larger-capacity drones. Note that the simulated and real-world drones operate similarly and therefore have the same reward function. As described in Appendix A.3.1 Line 737-740, "A real agent is an actual fire-fighting drone that captures the fire-image using its camera and acquires current sensor data using its on-board sensors to correspond with the captured image."  The fire-edges and hotspots are predicted, and changes in fire area and intensity result in the reward. A simulated agent's input is provided by the simulator. Other than the input source, though, the actual action space and rewards are the same for both real and simulated agents.
>
> 2. I'm not 100% certain … elaboration on that.
>
> The spread of model simulation fires was shown by the WRF-Fire model described in Section 4.1, Line 296 of the paper. The firefighter baseline comparing these fire scenarios was obtained through public datasets such as Singla et al. (2020), Fantineh (2023), Nguyen et al. (2024), Center (2025), and the NIFC dataset.
>
> For a specific target fire sample scenario, using the information from these datasets, we obtained additional data for the sample fire conditions, such as vegetation, crews, equipment involved, containment time, and percentages. This information was then correlated with the model to obtain the fire containment time and resources involved in fighting the fire. We will add further clarification in the final version of the paper.
>
> 3. Minor typos in the …  a few others.
>
> We sincerely apologize for issues regarding any typos in the paper. We will perform a very thorough proofreading of the entire paper to ensure a pristine write-up, clear of any issues with spelling, grammar, or any other typos.

---

> ### Author Response · Authors · 2025-12-03
> **Author Response #2 for Area Chair and Reviewer 19Yf**
>
> We thank the reviewer for their constructive feedback and appreciate their positive evaluation reflected in the initial score of 8/10. In response to their suggestions, we addressed their questions on agent activity assignments and shared policy, clarified details pertaining simulations and test baselines, and recommended editorial updates to the paper. These changes have strengthened the clarity and cohesiveness of the paper. We respectfully request the area chair to consider the reviewer’s positive assessment of the quality and sound basis of our work in the final decision.
>
> Based on the reviewer's comments, we have added clarifications and made refinements to the final version of the paper as described below:
>
> **Questions**:
> >1. "I am assuming that agents cannot be reassigned tasks… getting stuck in a loop if that is not the case."
>
> As disclosed in Section 3.1 and updates in Appendix A.3.2 [Lines 910-917]: "An agent is not required to operate on its assigned activity to completion. An agent's task assignment may be continuously … Upon completion of any activity, it can no longer be assigned to any agent."
>
> >2. "With the policy library being updated…question out of curiosity."
>
> We have fully addressed this in our Author Response #1.
>
> **Weaknesses**:
> >1. "I may have…this might help"
>
> We have fully addressed this in our Author Response #1.
>
> >2. "I'm not 100% certain about the baseline of the firefighters. I would love an elaboration on that."
>
> As clarified in update to Appendix A.3.2: [Lines 945-948]: "For a specific target fire sample scenario, using the information from the datasets, we obtained additional information related to the fire such as vegetation, containment time, and percentages. This information was then correlated with the model to obtain the fire containment time and resources involved in fighting the fire."
>
> >3. "Minor typos…I did pick up a few others"
>
> We have addressed the typos, grammar, and editorial corrections  in our final version of the paper.
>
> We thank this reviewer for their insightful comments and the area chair for their consideration.

---

### Meta-Review · Area_Chair_XmUh · 2026-01-07

**Summary:**

The paper proposes a "Scalable Multi-Agent Autonomous Learning" framework utilizing a two-phase iterative approach: a "Refocus" phase that assigns tasks using heuristics (A*, Transformers), and a "Refine" phase where agents execute policies and update them via "Shared Experience Learning." The authors demonstrate this system on a large-scale forest firefighting simulator with thousands of agents and provide a physical proof-of-concept.

However, I recommend a **Rejection** for this paper, after careful consideration of the paper's core novelty and writing quality. While the majority of reviewers (19Yf, ZeLM, 2oXC) were impressed by the system's scalability and the authors' responsiveness (specifically adding SMACv2 benchmarks), I find that the underlying algorithmic contribution is thin. The "novel" mechanisms proposed are largely re-implementations of standard techniques:
1.  **Phase 2 (Refine):** The "Shared Experience Learning" mechanism, described as a key innovation, appears to be a standard application of off-policy reinforcement learning using a shared replay buffer. The strategy of sampling trajectories (Algorithm 2 and 4) is mathematically equivalent to standard off-policy updates with a rudimentary form of Prioritized Experience Replay (PER).
2.  **Phase 1 (Refocus):** This phase relies heavily on domain-specific heuristics, such as A* planning and specific vision pipelines. While effective for the chosen application, this approach effectively bypasses the core challenge of multi-agent coordination by reducing the "learning" component to a local optimization problem. This limits the generality of the framework and its contribution to the MARL community.

And the methodological novelty must be better defined. The proposed "Refocus and Refine" structure bears significant resemblance to existing Hierarchical Reinforcement Learning (HRL) and dynamic task allocation frameworks, yet the submission fails to cite or compare against relevant prior art. The authors are required to discuss and differentiate their approach from similar works, such as [a,b]

Moreover, the manuscript currently suffers from conceptual ambiguities in the formal setup (Section 3.1) and a core illustration (Figure 1) that lacks clarity in depicting the two-phase process. The Captions for most of the Figures and Tables are too short to be self-contained. The authors are required to thoroughly proofread and revise the text to meet conference standards.

In summary, while the authors have demonstrated an impressive engineering effort capable of handling large-scale simulations, the paper does not meet the bar for acceptance due to a lack of fundamental algorithmic novelty and insufficient scholarly rigor. The reliance on standard off-policy updates and domain-specific heuristics suggests that the scalability is achieved through engineering constraints rather than learning breakthroughs. Consequently, the empirical results, while strong, cannot compensate for the limited methodological contribution and the failure to adequately situate the work within the broader context of HRL and task allocation literature. Therefore, I believe Rejection is the appropriate decision.


References:
- [a] Hierarchical attention master–slave for heterogeneous multi-agent reinforcement learning, Neural Networks 2023.
- [b] Learning Multi-Agent Coordination for Enhancing Target Coverage in Directional Sensor Networks, NeurIPS 2020.

**Reviewer Concerns:**

### Concerns Addressed by Rebuttal:
- Lack of Standard Benchmarks (Reviewers gFNh, ZeLM): The most critical weakness—missing comparisons on standard benchmarks—was addressed. In the rebuttal (Appendix A.5), the authors provided a detailed comparison against MAPPO and QMIX on SMACv2, demonstrating superior performance. This directly satisfied the primary condition set by Reviewer gFNh for acceptance.
- Theoretical Assumptions (Reviewer 2oXC): The authors clarified that assumptions of homogeneity apply to subgroups rather than the global population and added a "Practical Considerations" section to justify these assumptions in real-world contexts.

### Outstanding Concerns & The Basis for Rejection:
*   **Methodological Novelty (Aligned with Reviewer gFNh):** Reviewer gFNh initially characterized the work as "Limited Methodological Novelty," noting it reads like a "technical report." Although the reviewer mentioned they would accept the paper *if* benchmarks were added, I believe the addition of benchmarks does not cure the fundamental lack of novelty.
    *   **Critique of "Shared Experience Learning":** The authors frame the merging of trajectories from homogeneous agents as a novel contribution. However, analyzing **Algorithm 2 and Algorithm 4** (Appendix), this process is essentially collecting experience tuples $(s, a, r, s')$ into a central buffer ($D_{shared}$) and performing standard policy updates. The selection of "best/worst" trajectories is a heuristic variation of high-reward prioritization, which is well-explored in RL literature (e.g., Prioritized Experience Replay, Hindsight Experience Replay). The paper fails to distinguish why this specific instantiation is a novel *algorithmic* contribution rather than a standard implementation decision.
*   **Framework Generality (Aligned with Reviewer 2oXC):** The reliance on A* and domain-specific transformers in Phase 1 suggests the scalability comes from *engineering constraints* rather than *learning efficiency*. The framework essentially bypasses the hard MARL problems (credit assignment over long horizons) by using a hand-coded high-level planner. This limits the contribution to the specific domain of firefighting or spatially decomposable tasks.
*   **Missing Literature:** As pointed out by Reviewer gFNh and acknowledged in the rebuttal, the paper missed significant related work in Hierarchical RL and Dynamic Task Allocation. The proposed "Refocus/Refine" structure is a standard HRL pattern (Manager/Worker or Task Allocation/Execution), yet the paper does not adequately position itself against these established hierarchies.

**Reviewer Scores:**

Reviewer 19Yf (8->8): Positive evaluation; likely would remain 8.
Reviewer ZeLM (6->8): Engaged in the rebuttal and explicitly raised their score to 8 after seeing the SMACv2 results and clarifications.
Reviewer 2oXC (6->4/6): Found the framework clear but had theoretical concerns. With the clarifications on assumptions, this reviewer would likely have maintained a 6 or changed the score to 4.
Reviewer gFNh (4->4): While this reviewer offered a conditional acceptance based on benchmarks, the initial assessment ("Limited Methodological Novelty") was the most accurate. I am effectively upholding the original critique. The benchmarks show the *system* works, but they do not prove the *algorithm* is novel. Their score would likely remain low (**4**) if emphasizing the novelty aspect.

---

### Decision · Program_Chairs · 2026-01-26

Reject